# Customizing Spider Silk: Generative Models with Mechanical Property Conditioning for Protein Engineering

**Neeru Dubey**                                   *needub@kth.se*
*Division of Robotics, Perception and Learning*
*KTH Royal Institute of Technology*

**Elin Karlsson**                                 *elin.cb.karlsson@slu.se*
*Department of Animal Biosciences*
*Swedish University of Agricultural Sciences (SLU)*

**Miguel Angel Redondo**                          *miguel.angel.redondo@nbis.se*
*Science for Life Laboratory (SciLifeLab)*
*Uppsala University*

**Johan Reimegård**                               *johan.reimegard@scilifelab.se*
*Science for Life Laboratory (SciLifeLab)*
*Uppsala University*

**Anna Rising**                                   *Anna.Rising@slu.se*
*Department of Animal Biosciences*
*Swedish University of Agricultural Sciences (SLU)*

**Hedvig Kjellström**                             *hedvig@kth.se*
*Division of Robotics, Perception and Learning*
*KTH Royal Institute of Technology*

**Reviewed on OpenReview:** *https://openreview.net/forum?id=37YSapXDK6*

## Abstract

The remarkable mechanical properties of spider silk, including its tensile strength and extensibility, are primarily governed by the repeat regions of the proteins that constitute the fiber, the major ampullate spidroins (MaSps). However, establishing correlations between mechanical characteristics and repeat sequences remains challenging due to the intricate sequence–structure–function relationships of MaSps and the limited availability of annotated datasets. In this study, we present a novel computational framework for designing MaSp repeat sequences with customizable mechanical properties. To achieve this, we developed a lightweight GPT-based generative model by distilling the pre-trained ProtGPT2 protein language model. The distilled model was subjected to multi-level fine-tuning using curated subsets of the Spider Silkome dataset. Specifically, we adapted the model for MaSp repeat generation using 6,000 MaSp repeat sequences and further refined it via cross-validation on 592 repeats associated with experimentally determined fiber-level mechanical properties. Our model generates biologically plausible MaSp repeat regions tailored to specific mechanical properties, while also predicting those properties for given sequences. Validation includes sequence-level analysis, assessing physicochemical attributes, the expected distribution of key motifs, and secondary structure compositions. A correlation study using BLAST on the Spider Silkome dataset and a test set of MaSp repeats with known mechanical properties further confirmed the predictive accuracy of the model. This framework advances the rational design of spider silk-inspired biomaterials, offering a versatile tool for engineering protein sequences with tailored mechanical attributes.

# 1 Introduction

Recent advancements in protein design, particularly the integration of artificial intelligence (AI), have significantly enhanced our ability to engineer proteins with desired functions. Researchers have used deep learning techniques to improve the design of de novo proteins, achieving a tenfold increase in the success rates of target binding (Bennett et al., 2023). These innovations underscore the transformative potential of AI in protein engineering, paving the way for novel therapeutic interventions and biotechnological applications (Khakzad et al., 2023).

In parallel, the growing demand for sustainable, non-petroleum-based fibers has intensified interest in bio-derived alternatives. Spider silk, known for its exceptional mechanical properties and biodegradability, presents a promising candidate. However, efforts to develop artificial spider silk are hindered by limited knowledge of how the amino acid sequence of spider silk proteins (spidroins) influences the mechanical properties of the fibers. In this context, AI-driven protein engineering offers a powerful tool for designing spidroins that can be spun into fibers with customized performance characteristics.

Spiders spin up to seven different silk types, that all are composed of spidroins (Peakall, 1969). In this work, we focus on major ampullate silk, or dragline silk, renowned for exceptional mechanical properties - tensile strength comparable to steel (up to 1.3 GPa) with extensibility rivaling rubber (>30%) (Vollrath & Knight, 2001). This unique combination yields toughness exceeding both steel and Kevlar (Gosline et al., 1999), making spider silk particularly attractive for applications ranging from biomedical sutures to high-performance textiles (Bourzac, 2015).

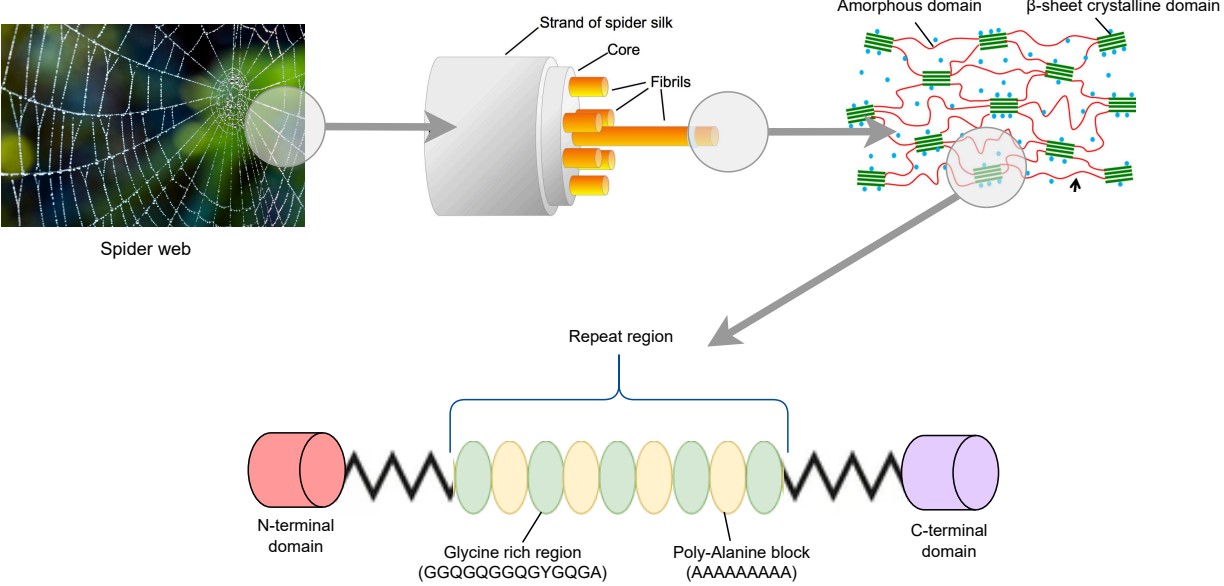

Figure 1: Hierarchical representation of the spider dragline silk fiber architecture, highlighting the schematic image of MaSp showing various sequential elements.

The molecular structure of MaSps comprises three primary regions: the globularly folded N-terminal and C-terminal domains, which are flanking the repetitive core region (Figure 1). The repetitive core region is believed to be the main contributor to the fiber mechanical properties, and in MaSps, this region is generally made up of poly-Ala blocks that are alternated with Gly-rich regions (Guerette et al., 1996). In the polymerized form, in the fiber, the poly-Ala blocks are predominantly arranged in nano-sized $\beta$-sheet crystals structures that contribute to the tensile strength of the silk fiber (Hijirida et al., 1996; Lewis, 1992;

Yarger et al., 2018b). The poly-Ala $\beta$-sheet crystals are embedded in an amorphous matrix formed by the Gly-rich repeats, which is related to the extensibility of the fiber (Gosline et al., 1984). Although often referred to as amorphous, the Gly-rich repeats also form specific conformations like $3_1$-helices, $\beta$-turns and $\beta$-spirals (Hayashi & Lewis, 1998; Hinman & Lewis, 1992; Van Beek et al., 2002).

Native major ampullate silk fibers from different spider species display large variability in mechanical properties. To elucidate the source of this large variability, Arakawa et al. (2022). undertook the significant challenge of sequencing the transcriptome of 1098 species and simultaneously determining the mechanical properties of major ampullate silks from 446 of these species. The results showed that there is a large interspecies difference in terms of mechanical properties. For example, the tensile strength of the major ampullate silks varies between 0.17 and 3.3 GPa (Arakawa et al., 2022). However, strong correlations between the amino acid sequence motifs in the repeat regions of the MaSps and the fiber mechanical properties could not be found (c.f. section 2.1).

Efforts to produce artificial spider silk have largely focused on replicating natural silk spinning processes through a variety of engineered approaches (Ferruz & Höcker, 2022; Koeppel & Holland, 2017). These methods majorly involve expression of engineered mini-spidroins in a heterologous host and subsequent spinning using an artificial spinning device (Schmuck et al., 2024). Although these methods have significantly advanced synthetic production, they often fail to provide a scalable means to tailor silk properties for specific applications. This limitation further underscores the need for computational approaches to predict and design sequence–property relationships effectively.

We propose a multi-stage framework for (1) generating MaSp repeat sequences conditioned on mechanical properties, and (2) predicting properties from given repeats. In contrast to prior work focused on full-length proteins (Lu et al., 2024), our model targets the repetitive core regions of MaSp proteins, which are central to fiber mechanics. The approach begins with distillation of ProtGPT2 (Ferruz et al., 2022) into a compact student model, *SpiderGPT*, trained on 100,000 spider proteins from UniRef50 (The UniProt Consortium, 2024). SpiderGPT is then fine-tuned in two stages using the Spider Silkome dataset (Arakawa et al., 2022): Level 1 uses 6,000 MaSp repeats to learn motif patterns, while Level 2 applies 5-fold cross-validation on 592 annotated sequences to learn sequence–property relationships. This hierarchical setup enables SpiderGPT to model both generative and predictive tasks with domain specificity and generalization.

To evaluate the model's effectiveness, we employed two distinct datasets: a test set of 185 instances sampled through cross-validation from the original 592-instance Spider Silkome dataset to assess self-consistency, and a BLAST set curated to determine sequence novelty against a broader protein sequence database. Our evaluation followed a comprehensive two-level analysis approach. At the sequence level, we analyzed key properties including molecular weight, instability index, isoelectric point, and the distribution of essential motifs (GGX, poly-Ala, YGQGG, and SV). For structural validation, we employed a secondary structure prediction tool to verify the composition of $\alpha$-helices, $\beta$-strands, and random coils characteristic of MaSps.

The model's ability to estimate the mechanical properties of silk fibers for a given MaSp repeat was evaluated by correlating generated and reference properties on a test set. The analysis yielded a cosine similarity of 0.9465 between the trend curves of generated and reference properties, indicating a strong alignment in predictive performance.

By combining generative modeling with biological validation, this work offers a robust computational framework for designing MaSp sequences with customizable mechanical properties. The findings hold promise for the advancement of synthetic biomaterial development and pave the way for applications in medicine, textiles, and engineering.

Our main contributions include:

- A novel computational framework that integrates knowledge distillation and multi-level fine-tuning to generate MaSp repeat sequences with targeted mechanical properties, addressing the challenge of limited mechanical property data.

- A dual-purpose model capable of both generating MaSp repeats based on desired mechanical properties and predicting mechanical properties from given sequences, offering flexibility for both design and analysis tasks.

- A comprehensive validation methodology that combines sequence-level analysis, structural prediction, and mechanical property correlation to ensure biological plausibility and functional relevance of generated sequences.

- Empirical demonstration of the model's effectiveness through statistically significant correlations between predicted and reference properties (cosine similarity of 0.94), while maintaining key structural motifs characteristic of spider silk proteins.

- A potential solution for the synthesis of sustainable biomaterials by providing a scalable approach for designing synthetic spider silk proteins with customizable mechanical properties.

The remainder of this paper is organized as follows: Section 2 presents a comprehensive review of the literature that covers spider silk sequence-property relationships, protein design using machine learning, and current approaches to modeling mechanical properties of spider silk. Section 3 describes our proposed methodology, including the model architecture and multi-level fine-tuning strategy. Section 4 presents our experimental setup, dataset details, and training procedures. Section 5 discusses our results, including self-consistency assessment, sequence generation analysis, mechanical property predictions, and ablation studies. Finally, Section 6 concludes with a discussion of potential applications and future research directions.

## 2 Literature Review

### 2.1 Spider Silk: Sequence-property Relationships

Major ampullate silk is widely recognized as a benchmark for high-performance biomaterials due to its unique combination of tensile strength, extensibility, and toughness (Vollrath & Knight, 1999; Gosline et al., 1999). The fiber is primarily composed of major ampullate spidroins (MaSps), and to date, five distinct MaSp classes (MaSp1–MaSp5) have been identified (Schmuck et al., 2024). Assignment of a MaSp to a specific class is typically based on clustering of its terminal domains in phylogenetic analyses and the presence of characteristic amino acid motifs within the repeat region.

A typical MaSp sequence comprises three main components: an N-terminal domain, a repetitive core region, and a C-terminal domain, as illustrated in Figure 1. The terminal domains are highly conserved and shared between different spidroins that give rise to fibers of very diverse mechanical properties. This conservation suggests that terminal domains are primarily involved in spidroin storage and polymerization rather than in determining fiber mechanics. Both N- and C-terminal domains have been extensively characterized in prior work (Andersson et al., 2014; Askarieh et al., 2010; Eisoldt et al., 2012; Šede et al., 2022). To the best of our knowledge, direct interactions between terminal domains and the repeat region have not been reported. Moreover, evidence from the literature consistently supports the conclusion that the mechanical properties of spider silk fibers are governed predominantly by the repetitive core domains of the spidroins (Anton et al., 2017; Bowen et al., 2018; Keten & Buehler, 2010; Gosline et al., 1999; Kono et al., 2021a; Vollrath & Knight, 2001; Schmuck et al., 2024). Consequently, our design strategy focuses exclusively on the repeat region, targeting the sequence elements most directly responsible for material performance. While we do not exclude the possibility of terminal–repeat interactions, our methodological emphasis remains on the core motifs most relevant to mechanical function.

Analyses of spidroin sequences in the Spider Silkome database have revealed several recurring amino acid motifs within the repeat regions of MaSps (Arakawa et al., 2022). Notable motifs include poly-Ala, GGX, YGQGG, SV, GPGXX, QQ, and AGQG (Arakawa et al., 2022; Keten & Buehler, 2010; Malay et al., 2017; Craig et al., 2020; Nakamura et al., 2024b; Kono et al., 2021b). However, attempts to correlate individual motif occurrences with fiber mechanical properties have yielded weak or no relationships, with Pearson correlation coefficients generally below 0.6 (Arakawa et al., 2022). Among the few exceptions are the YGQGG motif, which shows a positive association with toughness, and the SV motif, which is negatively correlated

with this parameter. Other frequently occurring motifs such as poly-Ala, GGX, QQ, and AGQG do not show strong individual correlations with fiber mechanics (Arakawa et al., 2022). These findings underscore the complexity of the sequence–function relationship in MaSps, which likely arises from synergistic interactions among multiple motifs rather than from the presence of any single motif.

In this study, we leverage recent advances in artificial intelligence to better understand the intricate sequence-to-property relationships within MaSp repeat regions. Through data-driven modeling, we aim to uncover latent patterns that govern the mechanical performance of spider silk fibers.

## 2.2 Protein Design Using Machine Learning and Generative Models

Deep learning has increasingly become a driving force in protein design, with generative models enabling data-driven exploration of protein sequence space. Among these, language model-based architectures have gained particular prominence by framing protein sequences analogously to natural language. Models such as ProtGPT2 (Ferruz et al., 2022), ProGen (Madani et al., 2020), and ProtBERT (Brandes et al., 2022) leverage transformer-based architectures trained on large-scale protein databases to generate biologically plausible sequences and predict structural or functional features.

These approaches have demonstrated success in designing enzymes (Madani et al., 2020), antimicrobial peptides (Das et al., 2021), and antibodies (Ruffolo et al., 2021), often surpassing traditional directed evolution methods in speed and diversity. More recently, models such as ProteinGAN (Repecka et al., 2021) and models conditioned on property constraints (Nijkamp et al., 2023; Lu et al., 2024) have emphasized functional design by guiding sequence generation toward specific biochemical or biophysical properties.

Structure-informed models like AlphaFold2 (Jumper et al., 2021), ESMFold (Lin et al., 2022), and diffusion-based frameworks (Watson et al., 2023; Anand et al., 2022) have further enhanced generative design by incorporating folding and energy priors, thus supporting the generation of not only functional but also structurally stable proteins.

In the context of spider silk proteins (spidroins), which are characterized by repetitive motifs and mechanical performance, these generative frameworks offer promising capabilities. However, sequence design aligned with fiber-level mechanical properties—such as toughness or strength—remains underexplored. Our work addresses this gap by conditioning sequence generation on mechanical property profiles, advancing property-aware generative modeling for hierarchical and repetitive proteins such as MaSp repeats.

## 2.3 Modeling Mechanical Properties of Spider Silk

Studies investigating the relationship between protein sequence and mechanical properties have primarily relied on sequence analysis and molecular simulations. For instance, (Tokareva et al., 2013) highlighted the importance of GGX and poly-Ala motifs in achieving spider silk's extensibility and tensile properties. Secondary structure prediction tools like PSIPRED (McGuffin et al., 2000) have been widely used to predict $\alpha$-helices, $\beta$-strands and random coil arrangements,, which are critical for understanding silk mechanics (Buchan & Jones, 2019). However, these approaches are often retrospective and do not enable forward design of sequences with desired properties. Efforts to model mechanical properties using machine learning have shown promise. For instance, Kim et al. (2023) utilized neural networks to predict silk's mechanical properties based on amino acid composition. While such methods enhance understanding of sequence-property relationships, they lack the generative capability required for designing novel sequences. Recent advancements in hybrid AI techniques have shown promise in overcoming these limitations, particularly for engineering silk-like proteins with desired mechanical characteristics (Lu et al., 2024; Ni et al., 2023; Fazio et al., 2023). Notably, Lu et al. (2024) introduced a generative modeling approach for spidroin sequence design, enabling the creation of synthetic spider silk proteins tailored to target mechanical properties. However, a key limitation of this approach is its lack of emphasis on MaSp repeat regions, which are the primary determinants of silk's mechanical behavior.

### 2.4 Our Proposed Approach in Context

Existing studies have made significant progress in predicting and designing silk proteins; however, they often overlook the critical role of the repeat regions in MaSp that govern mechanical properties. In this study, we explicitly remove the terminal domains from MaSp sequences and focus our analysis solely on the repeat region.

Although predictive models have contributed to understanding sequence–property relationships, they typically lack generative capabilities. Conversely, existing generative approaches do not focus specifically on the functional repeats of MaSp. To bridge this gap, we introduce a multi-level fine-tuning strategy tailored for MaSp repeat generation. In the first stage, a specialized model is trained to learn the structural and compositional features of MaSp repeats, ensuring biological plausibility in generated sequences. In the second stage, we fine-tune the model on annotated sequences with mechanical property labels, establishing a direct correlation between sequence motifs and their functional outcomes. This enables controlled generation of MaSp repeats with tunable mechanical characteristics.

By integrating domain-specific fine-tuning with generative modeling, our method extends beyond existing approaches, providing a more precise and biologically grounded framework for spider silk protein design. The ability to design spider silk proteins with tunable mechanical properties has significant implications for sustainable material innovation. Applications include eco-friendly, biodegradable textiles and high-performance biomedical materials such as tissue scaffolds and drug delivery systems (De Giorgio et al., 2024). More broadly, the combination of computational and experimental tools in protein engineering promises to accelerate the discovery of novel biomaterials and advance the field of biomimetic material design.

## 3 Proposed Pipeline: A Multi-Stage Fine-Tuning Framework

The target dataset, which maps MaSp repeats to their mechanical properties, is extremely limited in size, presenting a major challenge for training generative deep neural networks. To address this limitation, we adopt a multi-stage fine-tuning strategy that enables the model to perform two complementary tasks: (1) generating novel MaSp repeat sequences conditioned on desired mechanical properties, and (2) predicting mechanical properties from a given MaSp repeat.

The training pipeline is organized into three stages, as outlined in Figure 2, and is designed to progressively distill and specialize domain knowledge across tasks and data scales. While this hierarchical strategy is tailored to protein design, similar multi-stage or two-step post-training approaches have been explored in other domains. For instance, a recent study in audio modeling applies a conceptually similar two-phase training scheme for waveform generation (Sinha et al., 2024), described as a "two-step post-training" process.

The following sections detail the implementation and rationale of each stage within our framework.

### 3.1 Stage 1: Distillation of ProtGPT2 on Spider Sequences

ProtGPT2 (Ferruz et al., 2022)stands out among generative protein language models (PLMs) due to its specialized training and demonstrated ability to generate biologically plausible protein sequences. Unlike large-scale PLMs such as ESM-2 (Lin et al., 2023) and ProteinMPNN (Dauparas et al., 2022), which focus on structure prediction and sequence design constrained by existing protein scaffolds, ProtGPT2 is designed specifically for de novo protein generation. It has been trained on 10 million protein sequences from the UniRef50 dataset of the UniProt Knowledgebase (UniProtKB) (The UniProt Consortium, 2024) using a self-supervised learning approach. This extensive pre-training enables ProtGPT2 to predict the next amino acid in a sequence, effectively capturing the underlying "grammar" of protein structures.

However, while ProtGPT2 is a powerful model, it is not ideally suited for specialized tasks such as MaSp repeat generation, which requires the model to capture specific sequence patterns unique to spider silk proteins.

To address this challenge, we employ a multilevel strategy that leverages a pre-trained protein language model while optimizing it for our specific task. We begin with ProtGPT2. To enhance efficiency and adapt

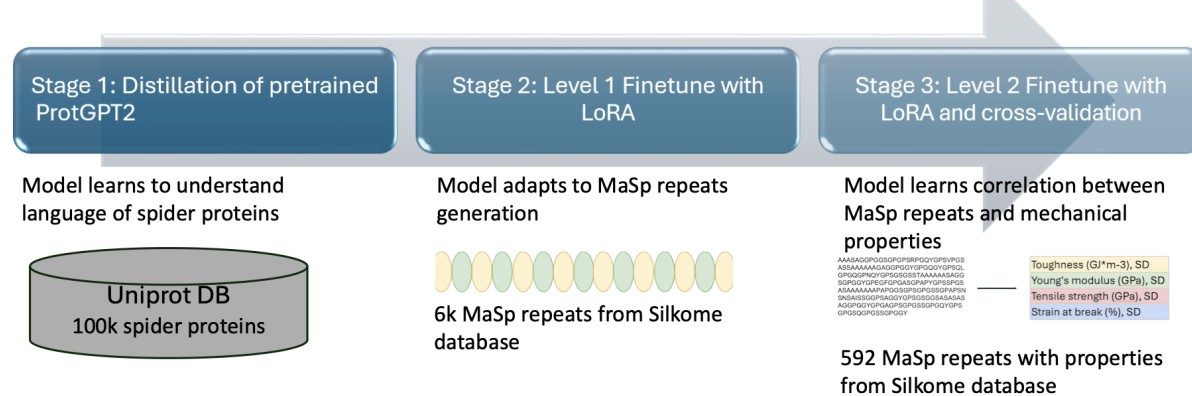

Figure 2: Illustration of the proposed methodology organized into three levels. Stage 1 involves training a distilled ProtGPT2 model using spider protein sequences from UniProtKB (The UniProt Consortium, 2024). In Stage 2, the model is fine-tuned on the repeat regions of MaSp to adapt to their unique patterns. Finally, Stage 3 further fine-tunes the model to capture correlations between MaSp repeats and their mechanical properties.

the model for MaSp repeat generation, we apply knowledge distillation (Hinton, 2015), creating a smaller, task-specific variant: SpiderGPT. This distilled model retains the essential knowledge of its teacher while significantly reducing model size and improving inference time, making it more practical for generating novel MaSp repeat sequences tailored to specific mechanical properties.

**Dataset**: SpiderGPT was trained on a curated dataset of approximately 100k protein sequences obtained from UniProtKB. These sequences were specifically selected based on their taxonomic classification within *Araneae* (spiders) and an annotation score greater than 1. (Details of the procedure are provided in Appendix A.1.) This focused dataset enables SpiderGPT to specialize in spider proteins while leveraging the broader knowledge distilled from the teacher model.

**Training Process**: The distillation process follows a standard teacher–student framework:

- **Teacher:** ProtGPT2, a large pre-trained protein language model, serves as the source of knowledge.

- **Student:** SpiderGPT, a lightweight model trained to replicate the teacher's behavior while maintaining computational efficiency.

Distillation was guided by several critical hyperparameters. The temperature parameter $T = 10$ was used to soften the teacher's output probability distributions, enabling the student to learn finer inter-token dependencies. An interpolation coefficient $\alpha = 0.1$ controlled the trade-off between soft targets from the teacher and hard targets from the original training data, promoting both generalization and retention of ground-truth patterns.

The SpiderGPT model was implemented with an embedding dimension of $n_{\text{embd}} = 512$, six transformer layers ($n_{\text{layer}} = 6$), and eight self-attention heads ($n_{\text{heads}} = 8$). This reduced architecture was selected to balance representational capacity with training efficiency, allowing the student model to retain critical domain knowledge while being computationally tractable.

For effective knowledge transfer, both teacher and student models were trained on the same tokenized dataset. The teacher model generated soft labels—probability distributions over the vocabulary for each token in the input—which served as training targets for the student. Unlike hard labels, these soft targets convey distributional information about alternative token probabilities, allowing the student model to learn nuanced structural and contextual relationships embedded in the teacher's output.

Our knowledge distillation approach resulted in a substantial reduction in model size and computational demands, without compromising the quality of sequence generation. A comprehensive analysis of the distillation process and its effects is presented in Section 5.3.

### 3.2 Stage 2: Level 1 Fine-Tuning on MaSp Repeat Regions

The SpiderGPT model, distilled on a spider protein dataset, learns to generate spider silk proteins. In Stage 2, fine-tuning on MaSp repeats refines its ability to recognize and generate MaSp-specific motifs and structures while preserving core protein language knowledge. This methodology demonstrates a sophisticated approach to transfer learning in computational protein design, which bridges the understanding of the fundamental protein language with specialized structural insights.

**Dataset**: For this study, we utilized the Spider Silkome dataset (Arakawa et al., 2022), a comprehensive resource cataloging silk gene sequences from 1,098 spider species and measuring mechanical, thermal, structural, and hydration properties for 446 species. It highlights the role of MaSp paralogs (MaSp1–MaSp3) in high-performance silk and identifies key amino acid motifs contributing to silk properties. This dataset serves as an open platform for advancing biomaterial research and innovation.

For level 1 fine-tuning, we curated a dataset of MaSp sequences, resulting in 6,000 instances. To focus on the functional repeat regions, we removed the first 150 and last 115 amino acid residues, which correspond to the N-terminal and C-terminal domains, respectively (De Oliveira et al., 2024). The repeat sequences of the same MaSp type were then concatenated for each species to create a structured dataset.

Let this dataset be represented as a collection of unlabeled MaSp repeat sequences:

$$\mathcal{D}_1 = \{s_i \mid s_i \in \Sigma^*, \ i = 1, \ldots, 6000\}$$

where $\Sigma$ is the standard amino acid alphabet and $s_i$ denotes a variable-length MaSp repeat sequence over $\Sigma$. This dataset was used to train the model to learn token-level dependencies and contextual representations across natural MaSp repeats.

**Training Process**: The fine-tuning procedure adopted a causal language modeling (CLM) objective, which exploits the auto-regressive nature of the transformer architecture. This objective enables the model to predict each token in a sequence conditioned on its preceding tokens, providing a principled framework for learning sequential dependencies in protein sequences.

To address the challenges of fine-tuning on a limited dataset, we utilized *Low-Rank Adaptation* (LoRA), a parameter-efficient technique that introduces trainable low-rank matrices into the attention mechanisms of the transformer. Rather than updating all weights, LoRA injects additional parameters into selected layers, substantially reducing the number of trainable weights while preserving model expressiveness. Our implementation followed the standard configuration from Hugging Face's PEFT library. Specifically, we used a LoRA rank of $r = 16$, which defines the dimensionality of the inserted low-rank matrices. A higher rank allows for increased adaptation capacity at the cost of more parameters. The scaling factor was set to $\alpha = 32$, amplifying the contribution of the low-rank component before integration into the base model. To regularize training, a dropout rate of 0.1 was applied within the LoRA layers. Biases in the original model layers were left unmodified (`bias=none`), and a weight decay coefficient of 0.01 was used to further constrain parameter growth during optimization.

The model was configured with a maximum sequence length of 512 tokens to accommodate variability in input length while ensuring consistent tensor shapes during batching. The tokenizer was adapted from the base model and configured to replace padding tokens with end-of-sequence tokens, ensuring compatibility with the CLM objective.

Training optimization included a learning rate of $5 \times 10^{-4}$ to enable rapid convergence, a small per-device batch size of 4 to manage memory constraints, and 50 warm-up steps to stabilize learning in the initial epochs. Training proceeded for a maximum of 10 epochs, with early stopping enabled based on validation loss to prevent overfitting. The Hugging Face `Trainer` API was used for streamlined integration of training, logging, and checkpointing.

To manage storage and preserve high-performing checkpoints, we retained only the top two model states based on validation metrics. This strategy ensured the availability of optimal models for downstream use while maintaining computational efficiency.

By integrating LoRA with careful regularization, early stopping, and resource-aware optimization, we established a robust fine-tuning pipeline capable of leveraging limited protein sequence data to train powerful generative models. This methodology provides a flexible and scalable foundation for downstream applications in protein design and biomolecular sequence modeling.

### 3.3 Stage 3: Level 2 Fine-Tuning for Sequence-Property Associations

To enable effective learning of sequence–property relationships in the MaSp dataset, we conducted Level 2 fine-tuning using a parameter-efficient fine-tuning (PEFT) strategy based on *Low-Rank Adaptation* (LoRA). This phase was designed to balance model generalization and computational efficiency, while maintaining evaluation rigor through a custom cross-validation framework. The fine-tuning objective was bidirectional in nature: (i) to generate MaSp repeat sequences with specified mechanical properties, and (ii) to predict mechanical properties given an input MaSp repeat.

This dual-task setup improves the model's ability to learn robust mappings between sequence and function, significantly enhancing its utility in the rational design of synthetic spider silk proteins.

**Dataset:** For the second level of fine-tuning, we curated a dataset of MaSp (major ampullate spidroin) repeat sequences and their corresponding mechanical properties from the Spider Silkome database, covering 293 spider species. To ensure high-quality and consistent annotations, we filtered the dataset to include only sequences with complete mechanical property data and excluded repeated entries at the MaSp subtype level. This filtering process yielded a final set of 592 unique MaSp repeat sequences, suitable for modeling sequence–property relationships. Each data point comprises the amino acid sequence of a MaSp repeat, along with four mechanical properties: toughness, Young's modulus ($E$), tensile strength, and strain at break, along with their respective standard deviations.

Let this dataset be represented as a set of labeled samples:

$$\mathcal{D}_2 = \{(s_i, \mathbf{c}_i, t_i) \mid i = 1, \ldots, 592\}$$

where:

- $s_i \in \Sigma^*$ is a MaSp repeat sequence over the amino acid alphabet $\Sigma$,

- $\mathbf{c}_i \in [0, 1]^8$ is the normalized conditioning vector, defined as:

$$\mathbf{c}_i = [\text{Toughness}_i, \sigma_{\text{Toughness},i}, \text{Strength}_i, \sigma_{\text{Strength},i}, E_i, \sigma_{E,i}, \text{Strain}_i, \sigma_{\text{Strain},i}]$$

    Each property is paired with its corresponding standard deviation, and all values are scaled to the range $[0, 1]$ via Min–Max normalization.

- $t_i \in \{\texttt{<GenerateSequence>}, \texttt{<EstimateProperty>}\}$ is the task token prepended to the model input to specify the prediction direction.

The model learns both directions through task conditioning: predicting properties from sequences and generating sequences from target property profiles, using the full 8-dimensional vector $(\mathbf{p}_i, \boldsymbol{\sigma}_i)$ as conditioning input or target as appropriate.

To standardize the feature space and facilitate model convergence, all property values and standard deviations were scaled to the $[0, 1]$ range using Min–Max normalization.

To support bidirectional training, we introduced a task-specific token at the beginning of each input, explicitly indicating whether the model should perform sequence generation or property estimation. This ensures clear task conditioning within a unified modeling framework. The two tasks are defined as:

- **Forward Task (Property Estimation):**

$$\texttt{<EstimateProperty>} \parallel s_i \rightarrow \text{Model} \rightarrow \mathbf{c}_i$$

- **Reverse Task (Sequence Generation):**

$$\texttt{<GenerateSequence>} \parallel \mathbf{c}_i \rightarrow \text{Model} \rightarrow s_i$$

**Training Process:** To adapt the model for learning sequence–property relationships from limited annotated data, we reused the LoRA-based architecture introduced during Level 1 fine-tuning. This included injecting trainable low-rank matrices into the attention mechanisms and leveraging the same dropout, weight decay, and optimization settings to maintain regularization and efficiency.

The model was trained using a learning rate of $1 \times 10^{-5}$, with a batch size of 8 per device. We used 50 warm-up steps to stabilize early optimization and trained for up to 10 epochs. An early stopping mechanism was activated based on validation loss to avoid overfitting. Tokenization was performed using a custom tokenizer derived from the base model, with all sequences padded and truncated to a fixed maximum length of 512 tokens. The end-of-sequence token was used for padding to ensure consistency with the model's CLM objective.

**Cross-Validation:** To evaluate model generalization under multiple data partitions while maintaining high training efficiency, we implemented a custom cross-validation strategy. The dataset was first divided into 16 equal subsets, each containing 37 sequences. Rather than training on all 16 folds, we randomly selected 5 representative folds to serve as validation sets in independent training runs. In each run, one subset was held out for validation, while the remaining 555 sequences were used for training. This approach allows for robust estimation of performance variance while controlling for training resource demands.

Each run employed a causal language modeling objective, in which the model predicts the next token in a sequence given its preceding context. After each training cycle, the fine-tuned model and tokenizer were saved separately, and the corresponding validation set was exported for downstream analysis. This approach enabled isolated evaluation across folds and supported robust assessment of generalization performance.

The resulting models exhibit a strong ability to capture complex, non-linear relationships between MaSp repeat amino acid sequences and their associated mechanical properties. By leveraging advanced transformer architectures in combination with principled fine-tuning and rigorous cross-validation, this approach offers a promising computational framework for the rational design of synthetic spider silk proteins. It highlights the potential of machine learning to derive meaningful insights from limited experimental data and demonstrates a scalable, generalizable strategy for biomaterial discovery and engineering.

## 4 Experimentation

### 4.1 Model Architecture

The SpiderGPT model emerges as a strategically distilled version of the original ProtGPT2, representing a sophisticated approach to computational efficiency in protein sequence modeling. Developed through knowledge distillation, the model maintains core architectural principles while significantly reducing computational complexity. Architecturally, the SpiderGPT is designed with precise specifications that balance performance and efficiency. The model features an embedding dimension of 512, compared to the original model's 1280, and comprises 6 transformer layers against the original 36. This reduction is accompanied by a corresponding decrease in attention heads from 20 to 8, and a substantial reduction in total parameters from 738 million to approximately 50 million. The embedding layer continues to serve as a critical component, implementing a specialized protein sequence representation approach. By learning contextual representations of amino acid sequences, the model captures molecular structural information with a hidden dimension of 2048, enabling nuanced analysis of protein sequence characteristics while maintaining computational efficiency. The multi-head attention mechanism remains a key innovation, allowing parallel processing of sequence information. With 8 attention heads, the model can simultaneously analyze multiple sequence

aspects, facilitating complex feature extraction and providing comprehensive insights into protein sequence relationships. This strategic model compression demonstrates a sophisticated approach to machine learning in protein sequence analysis. By preserving the core learning capabilities of the original ProtGPT2 while significantly reducing computational overhead, the SpiderGPT model offers researchers a more accessible and efficient tool for exploring protein sequence complexities. The model represents a critical advancement in computational protein modeling, bridging the gap between comprehensive sequence analysis and practical computational constraints. Its design reflects a nuanced understanding of both machine learning techniques and the intricate nature of protein sequence structures.

## 4.2 Setup

Our experimental evaluation focused on two critical aspects of the SpiderGPT model's performance in MaSp protein sequence analysis. The comprehensive assessment aimed to validate the model's capabilities in generating biologically meaningful sequences and understanding the intricate relationships between protein sequences and mechanical properties.

The first phase of experimentation concentrated on assessing the quality and biological plausibility of generated protein sequences. We employed a multi-metric approach to rigorously evaluate the generated sequences. This evaluation involved analyzing key molecular characteristics, including sequence composition, amino acid distribution, and structural coherence.

The second experimental phase delved into the complex relationship between MaSp repeat sequences and their mechanical properties. We sought to establish correlations between specific sequence characteristics and mechanical attributes such as toughness and elastic modulus. By systematically mapping sequence features to mechanical properties, we aimed to uncover the underlying molecular determinants that influence the material performance of spider silk proteins.

The experimental design was carefully constructed to provide insight into the predictive capabilities of the model and its potential to advance our understanding of protein sequence-structure-property relationships. By combining computational modeling with rigorous statistical analysis, we aim to bridge the gap between molecular-level sequence information and macroscopic material performance.

## 4.3 Unconditional Sequence Generation: Self-consistency Assessment

To assess the model's capability to generate biologically plausible sequences, we unconditionally generated 100 MaSp repeats from each of the five independently trained models in Level 2 fine-tuning (a sample of 50 sequences is provided in Supporting Document SD1). This resulted in a total of 500 sequences generated without conditioning on mechanical property values. To evaluate their fidelity, we compared the distribution of key biochemical and structural properties against the 592 natural MaSp repeats used during Level 2 fine-tuning. The results, presented in Figure 3, quantitatively assess the similarity between generated and natural sequences, highlighting how well the model captures fundamental features of spidroin repeat regions.

The probability distribution of amino acid occurrences in both natural and generated sequences was examined using KL divergence, which quantifies how one probability distribution deviates from another. Expressed in bits, KL divergence represents the additional information required to encode a sequence based on the background amino acid distribution of MaSp repeats (Appendix A.3). Additionally, Hamming distance was used to assess diversity in the generated sequences by measuring the number of differing positions between sequences of equal length, providing insight into the variation and uniqueness of the generated sequences relative to natural MaSp repeats.

To further assess the biological plausibility of the generated sequences, we performed a comprehensive analysis of key physiochemical attributes, including sequence length, molecular weight, instability index, and isoelectric point (see Appendix A.2 for more details). These properties were computed using ProtParam (Gasteiger et al., 2005), a widely used tool for protein sequence analysis, implemented via a Python-based tool package to ensure accuracy and reproducibility. The results indicate that the generated sequences exhibit similar distributions across these physiochemical attributes, aligning closely with natural MaSp sequences.

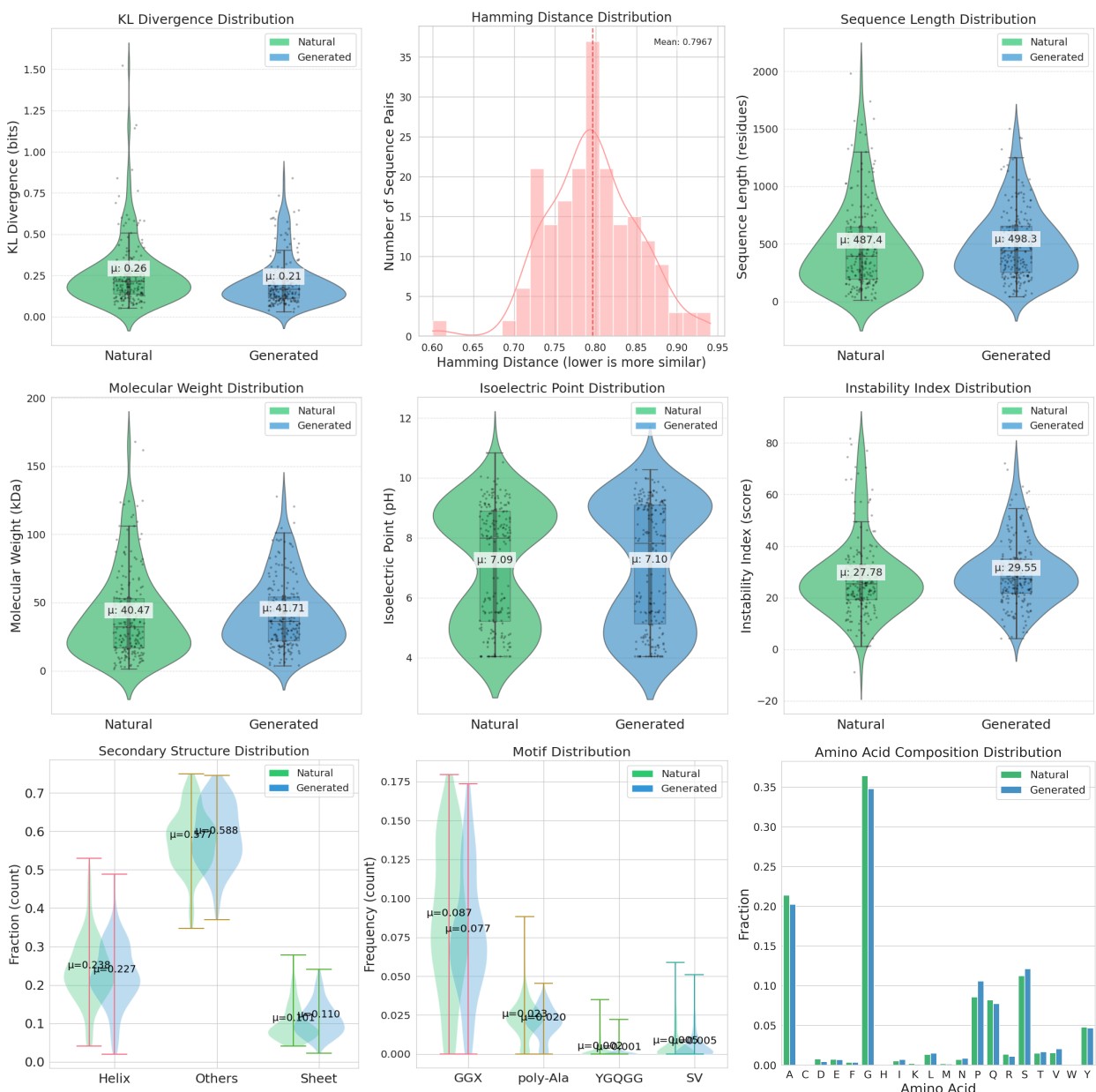

Figure 3: This figure presents a comprehensive comparison of nine key physicochemical and structural properties between naturally occurring (Natural) and computationally generated (Generated) proteins. The analysis includes distributions of KL divergence, Hamming distance, molecular weight, isoelectric point, instability index, sequence length, motif patterns, secondary structure elements, and amino acid composition. The plots demonstrate the degree of similarity between generated proteins and their natural counterparts across multiple biologically relevant parameters, providing insights into the fidelity of the protein design process.

Additionally, secondary structure fractions were analyzed using the ProteinAnalysis module from Biopython (Cock et al., 2009), which follows established secondary structure prediction frameworks, including DSSP (Kabsch & Sander, 1983) and Chou-Fasman propensity scales (Chou & Fasman, 1974). The ProteinAnalysis.secondary_structure_fraction() function was used to compute the fractional composition of $\alpha$-helices, strands and others/unstructured regions. This analysis enabled a quantitative comparison of secondary

structure content between natural and generated sequences, providing insights into structural stability and folding tendencies.

Furthermore, the frequency of key motifs in the generated sequences was examined as part of the secondary structure evaluation. Figure 3 presents the distribution of poly-Ala, GGX, YGQGG, and SV motifs, offering insight into their role in sequential integrity. The formal definition of these motifs is provided in Equation 1. The selected set of motifs are known to impact mechanical properties of the silk fiber (Arakawa et al., 2022; Nakamura et al., 2024b) (refer section 2.1). The motif frequency distribution of natural and generated sequences, illustrated in Figure 3, suggests that the generated sequences reflect a similar distribution as natural sequences.

To further explore the structural properties of the generated sequences, we analyzed the amino acid composition and secondary structure fractions, which are critical determinants of protein function and mechanical behavior. Figures 3 present these comparisons in detail. It's worth mentioning that the standard secondary structure prediction methods are optimized for globular proteins, making them less reliable for structural proteins like MaSp repeats (Jumper et al., 2021).

### 4.4 MaSp Motif-Property Correlation Analysis

The relationship between MaSp motifs and mechanical properties in spider dragline silk offers critical insights into how specific protein sequence patterns influence silk performance. Recent high-throughput studies have quantified key mechanical metrics -including toughness, Young's modulus, tensile strength, and strain at break - of dragline silk from various spider species, identifying correlations between motifs in the repetitive core region of MaSps and variations in these properties (Arakawa et al., 2022).

Structural motifs such as poly-Ala (associated with $\beta$-sheet crystallization) and GGX (contributing to elastic $\beta$-turns) play a direct role in defining tensile strength, toughness, and extensibility. In addition to these well-characterized motifs, recent analyses have identified emerging motifs like YGQGG, QQ, GPGXX, AGQG and SV, which may further modulate mechanical behavior (refer section 2.1). The regular expression of each motif, which we used for current analysis, is mentioned in equation 1. Understanding these relationships is instrumental in the design of biomimetic silk materials with tailored mechanical properties.

$$\text{motifs} = \begin{cases} \text{YGQGG:} & YGQGG \\ \text{poly-Ala:} & A\{3,\} \\ \text{GGX:} & GG[A-Z] \\ \text{QQ:} & QQ \\ \text{GPGXX:} & GPG[A-Z]\{2\} \\ \text{AGQG:} & AGQG \\ \text{SV:} & SV \end{cases} \tag{1}$$

In previous studies, correlation analyses have primarily focused on the distinct motifs present in different MaSp types (MaSp1, MaSp2, and MaSp3) and their specific influences on mechanical properties (Craig et al., 2020; Nakamura et al., 2024a; Kono et al., 2021a; Arakawa et al., 2022). By contrast, the current work considers MaSp more abstractly, providing an overarching view of how various motifs may affect mechanical performance.

For the correlation study, we refined the level 2 fine-tuning dataset by selecting instances with unique sets of mechanical properties. This resulted in a filtered subset of 294 instances. Using this refined dataset, we conducted a reverse generation task, where the model generated 294 MaSp repeats corresponding to these specific property sets. The generated sequences were then used for correlation analysis, allowing a structured evaluation of the relationship between protein motifs and mechanical properties. This study aims to determine whether the generated sequences accurately reflect the sequence-to-property correlations observed in previous research.

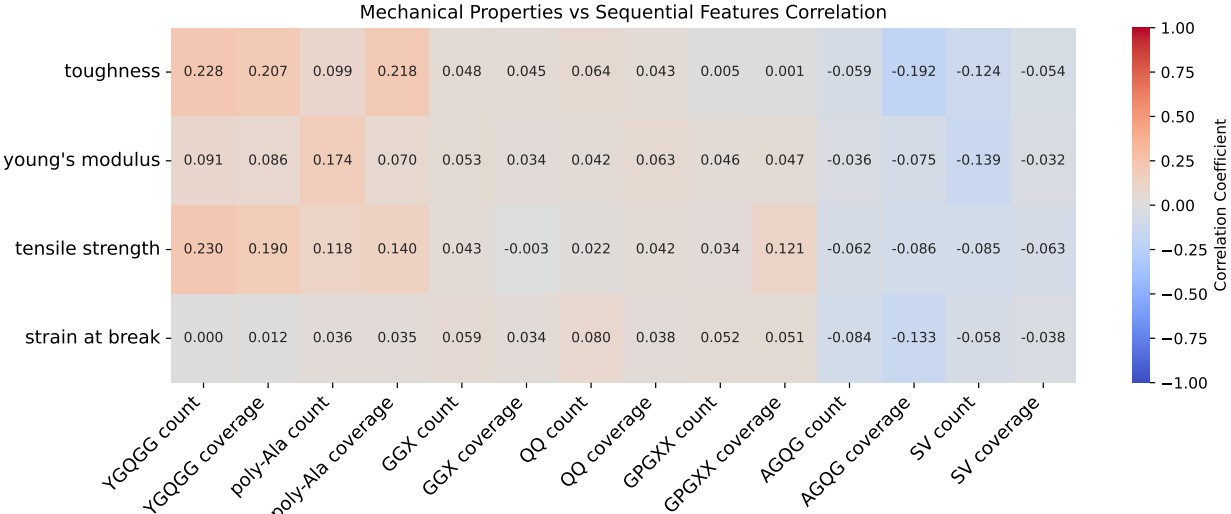

Figure 4: The heatmap illustrates the correlation between sequential features and the mechanical properties of the generated sequences. While some weak correlations are present, the overall low values suggest the need for a more advanced approach to better capture sequence-property relationships.

Figure 4 presents the correlation heatmap between sequential features and mechanical properties. The sequential features include the count and coverage ($\frac{\text{motif size} \times \text{count}}{\text{sequence length}}$) of the specified motifs. The heatmap indicates that correlation values remain relatively low. While no strong correlations were observed, this visualization is still informative, demonstrating that individual motifs alone cannot fully explain the relationship between sequence variation and mechanical properties. This highlights the intricate and multi-factorial nature of silk mechanics, where the combined influence of multiple motifs and structural contexts collectively shapes the observed mechanical behavior.

In the following, we present a row-wise analysis of the heatmap:

### Toughness (Energy to Break)

Toughness is the total energy absorbed before failure. Higher toughness in dragline silk is associated with motifs that enhance tensile strength and/or extensibility. The correlation heatmap reveals that YGQGG count (+0.228) and coverage (+0.207) both show moderate positive correlations with toughness. Poly-Ala coverage also correlates positively with toughness, though slightly less strongly (+0.218). The AGQG coverage (-0.192) and SV count (-0.124) showed negative correlation with toughness.

### Young's Modulus (Stiffness)

Young's modulus represents the initial stiffness of silk fibers and is influenced by motifs that foster $\beta$-sheet crystallinity. For example, poly-Ala count correlates positively with Young's modulus (+0.174). In our analysis, we also observed a positive impact of the count of YGQGG motif on Young's modulus (+0.091), whereas the count of AGQG (-0.036) and SV (-0.139) motifs showed negative correlations.

### Tensile Strength (Ultimate Stress)

Tensile strength is the maximum stress that the silk fiber can withstand before failure. The correlation heatmap indicates that many of the same motifs influencing toughness also affect strength. The YGQGG motif count is positively correlated with tensile strength (+0.230) (Arakawa et al., 2022), making it one of the strongest sequence predictors of a stronger fiber in our analysis. Poly-Ala coverage (+0.140) and GPGXX coverage also positively correlate with tensile strength.

**Strain at Break (Extensibility)**

Strain at break is the maximal elongation (expressed as a fraction of its original length) that the silk fiber can achieve before breaking. The correlations highlight AGQG coverage (–0.133) is the most striking correlation overall (and the only moderately strong negative value). Other features in this row hover near zero, indicating minimal correlation. The results also show impact of QQ motif on strain at break. Other features in this row hover near zero, indicating minimal or no correlation.

In summary, the correlation between mechanical properties and individual motifs in the generated sequences follows patterns reported in previous studies (Arakawa et al., 2022; Nakamura et al., 2024b). However, the overall weak correlations suggest that individual motifs alone cannot account for the observed variations in fiber mechanical properties (Arakawa et al., 2022). This study builds upon existing sequence-property relationships and leverages machine learning to better capture the complex interplay between sequential motifs and mechanical properties, enabling the design of novel MaSp sequences.

## 5 Results and Discussions

The proposed framework performs both the forward task of generating MaSp repeats tailored to specific mechanical properties and the reverse task of predicting mechanical properties from a given MaSp repeat. This section presents the evaluations performed to assess the model's ability to generate biologically plausible sequences and accurately predict mechanical properties.

### 5.1 Biological plausibility of generated sequences

We perform an in-depth investigation using two datasets: the test set and the BLAST set. The test set consists of sequences sampled from the Spider Silkome dataset, allowing us to assess the model's self-consistency and its ability to reproduce meaningful sequences conditioned on known mechanical properties. The BLAST set, on the other hand, is used to evaluate the novelty and classification of the generated sequences within broader protein sequence databases. By integrating these evaluations, we aim to ensure that the generative model produces biologically plausible sequences that captures key features that are essential for functional spidroin design.

#### 5.1.1 Conditional Generation Assessment: Evaluation on test dataset

To assess the self-consistency of the trained model, we evaluated its performance on the test sets from all five models trained during Stage 3 (Level 2 fine-tuning). Each model was tested on a set of 37 instances, resulting in a total of 185 unique test records. Each instance includes a MaSp repeat sequence along with its corresponding mechanical property set. The detailed composition of the test set is provided in the supplementary document SD2. The mechanical properties from these test instances were used as input conditions for sequence generation.

We note that the sequence–property relationship is not strictly one-to-one. The comparison is based on the hypothesis that similar mechanical properties can arise from sequences that share structural or motif-level features, particularly within the MaSp repeat regions. The objective was to qualitatively assess whether the generated sequences resemble natural ones associated with similar mechanical profiles.

To evaluate how well the model captures the relationship between sequence and function, we compared the generated sequences to their natural counterparts at both the sequence and structural levels. As shown in Figure 5, this dual-level analysis highlights the model's ability to reproduce key biochemical and structural characteristics of spider silk proteins when conditioned on specific mechanical properties.

The sequence-level evaluation assesses key physiochemical attributes, including molecular weight, instability index, and isoelectric point as well as motif occurrences, including poly-Ala, GGX, YGQGG, and SV. Figure 5 illustrates the experimental results. The high agreement between these attributes in the generated and original sequences suggests that the model effectively learns to generate biologically plausible MaSp repeats when conditioned on a specific set of mechanical properties.

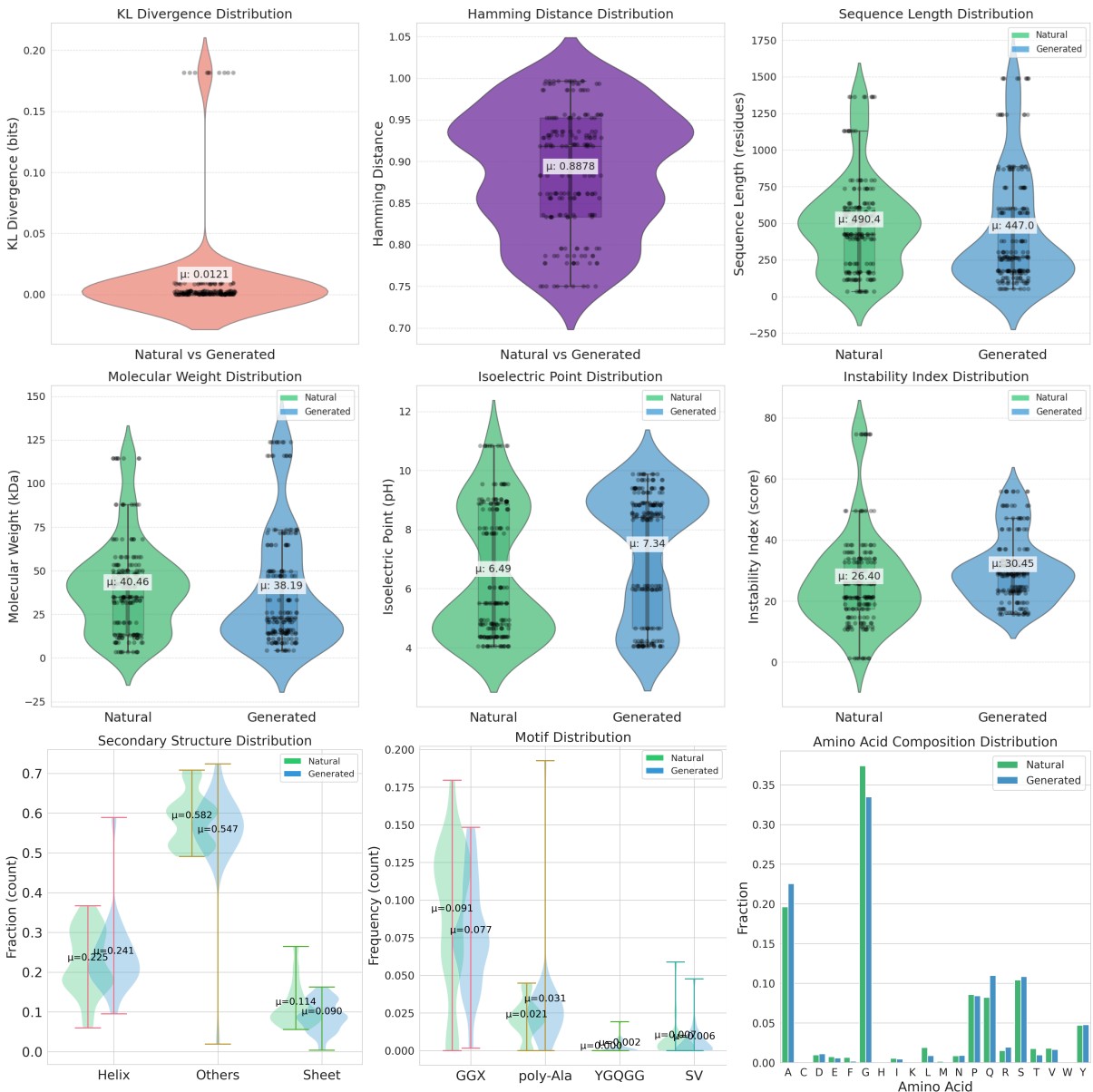

Figure 5: Comparison between original (natural) and generated sequences on the test set in terms of various matrices. (1) Sequence properties: sequence length, molecular weight, instability index, Isoelectric point. (2) Average amino acid frequency distribution grouped by physicochemical properties. The property consistency highlights the validity of the generated sequences in terms of their structural and biochemical features. Furthermore, the consistent alignment demonstrates the model's ability to effectively capture the key characteristics and properties of MaSp.

Additionally, sequence similarity was analyzed using KL divergence and Hamming distance. KL divergence quantifies the deviation in amino acid probability distributions between original and generated sequences. As shown in Figure 5, KL divergence remains constrained within a low range, indicating that the model successfully captures the natural amino acid distribution of MaSp repeats. In contrast, Hamming distance evaluates sequence diversity at the character level. The high Hamming distance values demonstrate the

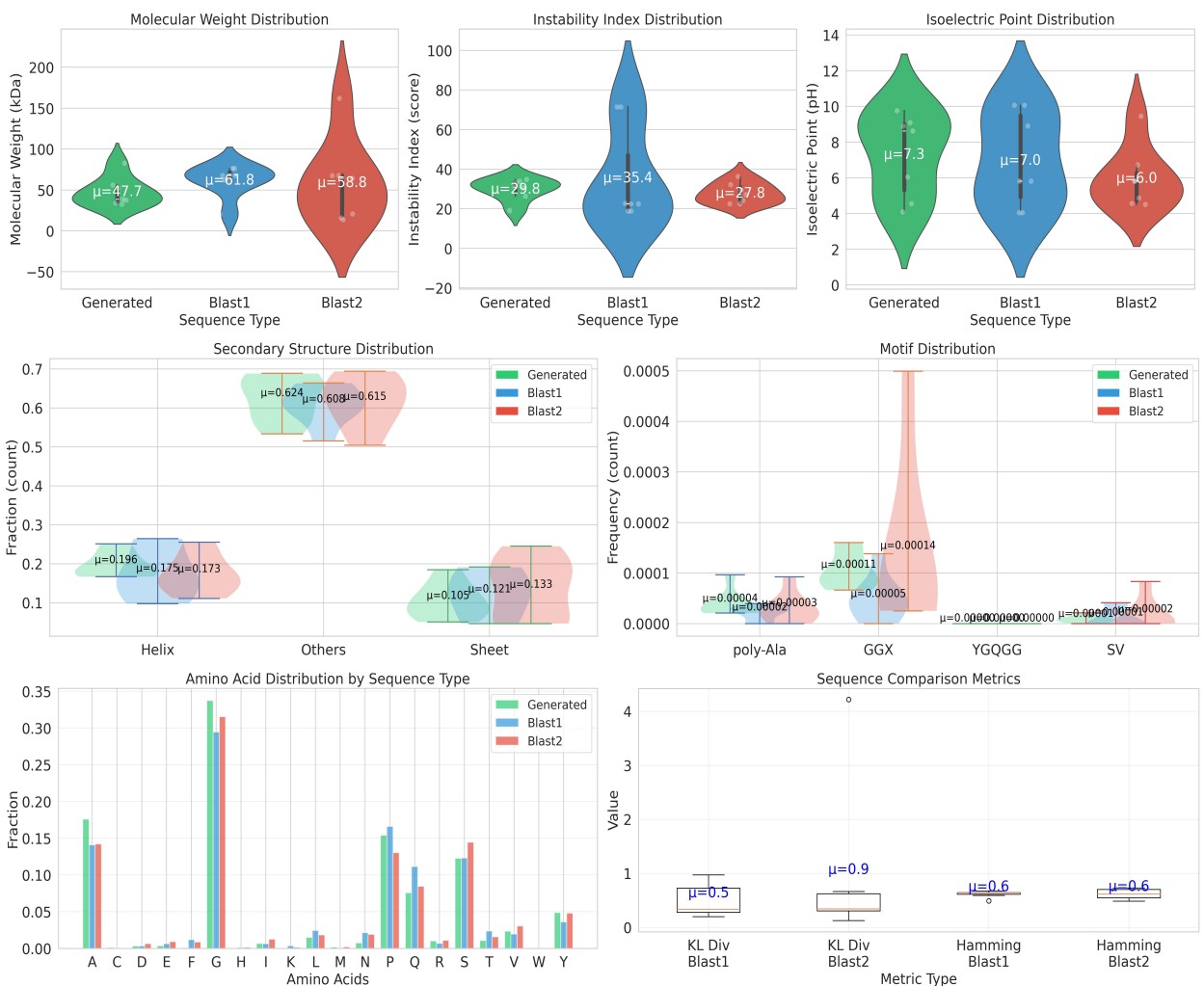

Figure 6: Comparison of sequence and structural properties of our generated protein sequences with BLAST1 and BLAST2 sequences. The analysis includes distributions of instability index, isoelectric points, molecular weight, motif prevalence, secondary structure fractions, amino acid composition, and sequence similarity metrics. This highlights the consistency of the generated sequences with natural proteins and their alignment across various properties

model's ability to generate diverse sequences, avoiding excessive similarity to training data while preserving essential biological patterns.

The structural evaluation further validates the sequence fidelity by analyzing key secondary structure features— $\alpha$-helices, $\beta$-strands and unstructured regions. The results indicate a strong structural resemblance of the secondary structure composition between generated and natural sequences, confirming that the model preserves critical biological motifs found in naturally occurring MaSp repeats (Figure 5).

The ability of the generative model to replicate both physiochemical and structural characteristics underscores its potential as a powerful tool for biomaterial design. By tailoring sequences to meet specific functional and mechanical property requirements, this approach offers a promising avenue for advancing the development of engineered biomaterials with enhanced properties.

### 5.1.2 Novelty Assessment: BLAST Evaluation

To evaluate the novelty of the generated MaSp repeats, we performed BLAST analyses against a broader Spider Silkome spidroin repeat database. The search space consists of 11K naturally occurring spidroin sequences. It includes seven primary types of spider silk proteins major ampullate spidroins (MaSp), flagelliform spidroins (Flag), minor ampullate spidroins (MiSp), aggregate spidroins (AgSp), pyriform spidroins (PySp), aciniform spidroins (AcSp), and cylindriform/tubuliform spidroins (CySp) (Yarger et al., 2018a). A subset of seven generated sequences was compared with the two most closely related natural sequences, designated BLAST1 (top match) and BLAST2 (second top match) based on BLAST results. The selection was based on high query coverage, percent identity, and sequence length similarity to the generated sequences. This dataset is referred to as the "BLAST set", with details of all sequences and their sources provided in the supporting document SD3. All selected BLAST matches had an expect value (E) of $E \leq e^{-10}$, which ensured statistically significant similarities. Furthermore, we selected matches spanning different MaSp subtypes (MaSp1, MaSp2, MaSp3) to provide a comprehensive comparative analysis. The expect value (E) quantifies the probability of obtaining a match by chance in a given database size, decreasing exponentially as the alignment score (S) increases.

To assess uniqueness, we follow established criteria where sequences with similarity values below 50–60% are considered novel (Quan et al., 2023). The generated sequences meet this threshold when compared to broader protein databases, demonstrating the model's ability to design distinct sequences that either do not exist in nature or have not yet been observed. Additionally, BLAST search results consistently classify the most similar existing sequences as MaSp, aligning with the intended spidroin type. This indicates that the generative modeling approach effectively captures the defining characteristics of specific spider silk proteins.

Figure 6 presents the comparison between the generated MaSp repeats and the closest matches, BLAST1 and BLAST2. The observed consistency highlights the ability of the model to accurately reproduce essential sequence features of spider silk proteins. The generated sequences maintain comparable molecular weight ranges, predicted secondary structure composition, and physiochemical properties, all of which are crucial for mimicking the functional behavior of natural spider silks. Furthermore, the model effectively balances critical sequence attributes, such as the instability index and isoelectric point, reinforcing the biological plausibility of the generated sequences.

Although Figure 6 confirms that the generated sequences share similarities in fundamental composition, structural, and sequence pattern with naturally occurring MaSps, reinforcing their biological relevance, variations in metric values may arise due to the selection of reference sequences for the BLAST comparison. This highlights the sensitivity of the evaluation process to the chosen dataset. Despite these variations, the model consistently captures the essential properties of spider silk proteins, further emphasizing its potential for biomaterial design and the generation of novel sequences with customized functionalities.

Table 1: Quantitative comparison of mechanical property prediction accuracy between SpiderGPT (ours) and SilkomeGPT (baseline) based on trend similarity metrics computed over reference and generated mechanical property profiles. Bold values indicate better performance for each metric.

| Metric | SpiderGPT (ours) | SilkomeGPT (baseline) |
|---|---|---|
| Pearson Correlation ($r$) | **0.8884** | 0.8349 |
| Spearman Correlation ($\rho$) | **0.8343** | 0.7798 |
| Mean Absolute Error (MAE) | **0.0861** | 0.0963 |
| Root Mean Square Error (RMSE) | **0.1047** | 0.1162 |
| $R^2$ Score | **0.6383** | 0.5861 |
| Cosine Similarity | **0.9827** | 0.9783 |

### 5.2 Evaluation of Mechanical Property Prediction

To evaluate the model's ability to estimate mechanical properties from MaSp repeat sequences, we use the test set of 185 instances described earlier. For each sequence in this set, we perform inference using the

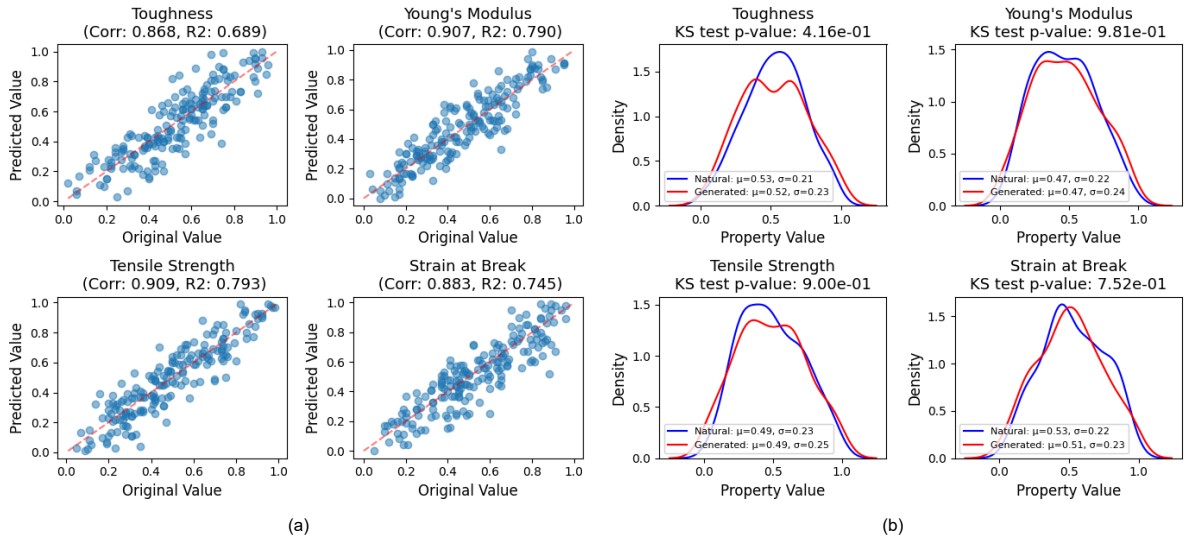

Figure 7: Comparison of predicted and original mechanical properties for MaSp repeats in the test set. (a) Scatter plots show the correlation between predicted and actual values for normalized mechanical properties, with Pearson's $r$ and ideal 1:1 (red dashed) line. (b) Kernel density plots compare natural and generated distributions for each property, including KS test $p$-values, means ($\mu$), and standard deviations ($\sigma$). Results indicate effective property prediction by the model.

model's reverse task—predicting mechanical properties from sequence input. The predicted properties are then compared against the corresponding ground-truth values.

We benchmark the performance of our model (SpiderGPT) against SilkomeGPT (Lu et al., 2024). Since SilkomeGPT does not focus exclusively on repeat regions, we use the original (unprocessed) sequences from the test set—that is, before repeat extraction—for both models to ensure a fair comparison in the mechanical property prediction task.

Prediction quality is assessed using five standard statistical metrics: Pearson's correlation coefficient ($r$), Spearman's rank correlation ($\rho$), mean absolute error (MAE), root mean square error (RMSE), and cosine similarity. The results, presented in Table 1, provide a quantitative comparison of the models' ability to replicate the mechanical property patterns in the reference data. It is important to note that SilkomeGPT was trained on the entire Spider Silkome dataset, meaning that the test instances used here were part of its training data. Despite this advantage, SpiderGPT—trained only on repeat regions and without exposure to the test set—outperforms SilkomeGPT across all evaluated metrics.

The Pearson correlation coefficient was found to be $r = 0.8884$, indicating a strong positive linear relationship between the predicted and reference values. This suggests that the model captures a substantial portion of the underlying variance in the data, even though the relationship is not perfectly linear. Additionally, the Spearman correlation coefficient was $\rho = 0.8343$, reflecting a strong monotonic relationship. Unlike Pearson, Spearman correlation assesses the consistency of the rank order between two variables. A high $\rho$ value implies that when a reference property increases, the corresponding predicted property also tends to increase, even if the absolute values do not match precisely. This highlights the model's ability to preserve the relative ordering of mechanical properties.

The MAE was measured to be 0.0861, indicating that, on average, predicted values differ only slightly from the reference values. The RMSE was slightly higher at 0.1047, consistent with its sensitivity to larger deviations. This implies that while most predictions are accurate, a few instances exhibit moderate error, which remains within acceptable bounds.

The coefficient of determination ($R^2$) was observed to be 0.6383. In simple linear regression, $R^2$ is equal to the square of the Pearson correlation coefficient, i.e., $R^2 = r^2$ Substituting $r = 0.89$, we obtain $R^2 = (0.89)^2 \approx 0.79$. However, the current dataset involves multiple output dimensions and includes non-linear dispersion, which can lower the observed $R^2$ despite a high average correlation. This explains the discrepancy between the theoretical and observed values, with $R^2 = 0.6383$ being a reasonable outcome in this context.

In addition, the cosine similarity between predicted and original property vectors was calculated to be 0.9827. This high value indicates strong directional alignment, implying that the overall structural trends are well preserved even if individual values are not perfectly matched.

A property-level analysis is provided in Figure 7. Panel (a) displays scatter plots comparing predicted and original values across the four mechanical properties: toughness, Young's modulus, tensile strength, and strain at break. Each subplot includes the Pearson correlation coefficient ($r$) and the coefficient of determination ($R^2$), offering a detailed view of model performance. All properties exhibit strong positive correlations, with tensile strength achieving the highest $R^2$ (0.793) and toughness the lowest (0.689), indicating robust but slightly variable predictive accuracy across property types.

Panel (b) of Figure 7 shows kernel density estimates comparing the distributions of natural and generated property values. Each subplot includes the Kolmogorov–Smirnov (K–S) test $p$-value, along with the mean ($\mu$) and standard deviation ($\sigma$) for both distributions. High $p$-values across all properties (e.g., 0.752 for strain at break) suggest that the generated properties are statistically consistent with the natural ones, as measured by the K–S test (Massey Jr, 1951).

## 5.3 Ablation Studies

In this section, we perform ablation experiments over a number of facets of the proposed methodology in order to better understand their relative importance.

### 5.3.1 Without Distillation

The first stage of the architecture pipeline involves distilling the ProtGPT2 model into a smaller SpiderGPT model. In this section, we explain the need for this technique and compare the ProtGPT2 and SpiderGPT models.

Pretrained protein language models (PLMs) like ProtGPT2 are large and designed for general protein generation tasks. While these models excel in broad protein generation, the current task is more specific. It involves generating MaSp repeats from a smaller dataset, where the sequences follow a distinct and specialized pattern. Given the focused nature of the task, a lighter model is more appropriate. It not only improves inference speed but also simplifies the overall architecture, making it more efficient for the specific requirements of this task.

Table 2 provides an architectural comparison between ProtGPT2 and SpiderGPT, highlighting that the SpiderGPT model is significantly more compact than its baseline counterpart, ProtGPT2. To assess their performance differences, we generated 200 protein sequences using each model. The reduced size of SpiderGPT not only simplifies the pipeline complexity but also accelerates inference, with the distilled SpiderGPT achieving a remarkable six-fold increase in inference speed compared to ProtGPT2. Evaluations conducted on these 200 sequences reveal that this boost in efficiency incurs only minimal performance trade-offs. Specifically, the student model, SpiderGPT, sustains perplexity levels comparable to those of the teacher model, ProtGPT2. For a more detailed visual comparison of the two models' performance, refer to Figure 8, which illustrates the outcomes of both the ProtGPT2 teacher model and the SpiderGPT model.

### 5.3.2 Without first level fine tuning

ProtGPT2 is a decoder-only transformer model that has been pre-trained on the protein space using the UniRef50 database (version 2021_04) (The UniProt Consortium, 2024), which contains 10 million protein instances. However, the Sequence-to-Mechanical Property Correlation dataset, curated from the Spider Silkome database, contains only 592 instances. This small dataset is insufficient for training the model

| Attribute | ProtGPT2 | SpiderGPT |
|---|---|---|
| Embedding Dim ($n_{embd}$) | 1280 | 512 |
| Layers ($n_{layer}$) | 36 | 6 |
| Hidden Dim | 5120 | 2048 |
| Attention Heads | 20 | 8 |
| Total Parameters | 738M | 50M |

Table 2: Comparison of Teacher and Student Model Architectures

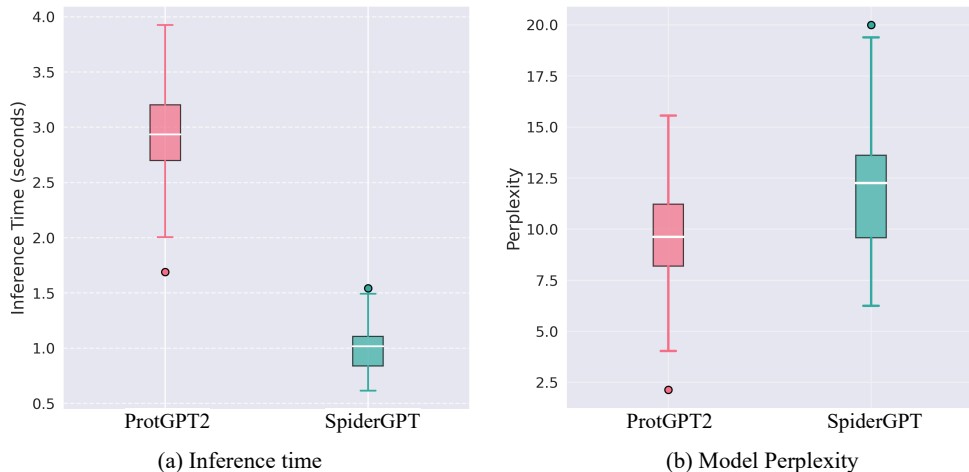

(a) Inference time                    (b) Model Perplexity

Figure 8: Comparison of performance of teacher (ProtGPT2) and student (SpiderGPT) models over generation of 200 novel protein sequences.

to generate MaSp repeat sequences while simultaneously learning the correlation between sequences and mechanical properties. To address this limitation, we split the training into two stages.

In the first stage, we focus on training the model to generate MaSp repeats. In the second stage, we train the model to learn the correlation between sequences and their corresponding mechanical properties. This staged approach allows the model to first specialize in generating MaSp repeats before learning to predict mechanical properties from these sequences.

In this section, we evaluate the impact of the first level of fine-tuning, where the model is trained on 6,000 instances of MaSp repeats. To emphasize the significance of this stage, we also present the model's performance when fine-tuned directly on the 592 instances from the Spider Silkome dataset, which contains MaSp repeats along with their corresponding mechanical properties. This comparison highlights the importance of multi-level fine-tuning in achieving optimal results.

For consistency, we used the same training setup across both scenarios. The SpiderGPT model was fine-tuned with the following LoRA settings: low-rank dimension ($r$) = 16, scaling factor ($\alpha$) = 32, dropout = 0.1, and weight decay = 0.01. Training was conducted for 10 epochs with a learning rate of $5 \times 10^{-4}$ and a per-device batch size of 4. Additionally, 50 warmup steps were used, and weight decay was applied at a rate of 0.01. Below we evaluate the impact of this ablation study on both forward (sequence generation) and reverse (property estimation) tasks.

**Impact on Sequence Generation Quality**

As discussed in previous sections, the primary objective of the model in the forward task is to generate MaSp repeats tailored to a given set of mechanical properties. In this section, we evaluate the impact of

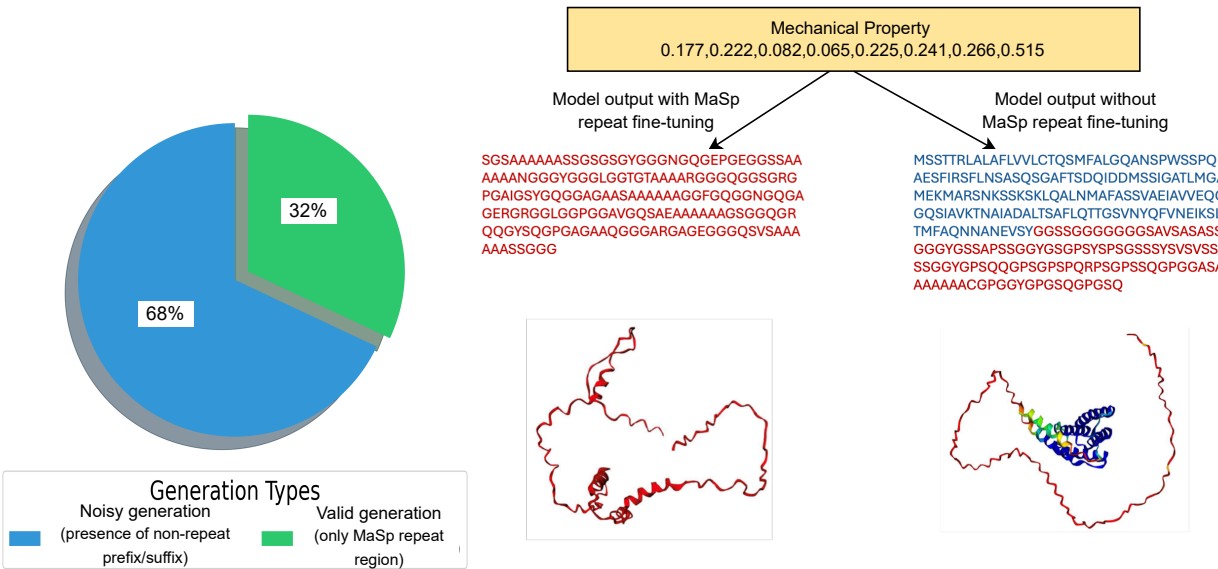

Figure 9: Impact of level 1 fine-tuning on sequence generation quality. Without MaSp fine-tuning, the model generates non-repeat prefix/suffix (68% occurrence), leading to structural deviations. The molecular structure predicted by Omegafold reveals helical formations, indicating the presence of terminal domains. In contrast, model achieves 100% valid sequence generation if we include MaSp repeat fine-tuning step in the methodology pipeline.

omitting the initial fine-tuning phase (level 1) on sequence generation quality. To this end, we generated 100 sequences using two models: one trained with both levels of fine-tuning and another trained without level 1 fine-tuning.

The omission of level 1 fine-tuning resulted in significant deviations in sequence generation, with the model frequently producing spider silk sequences containing non-repeat regions—an unintended outcome. To detect these deviations, a sliding window algorithm was employed to analyze the density of poly-alanine and glycine-rich regions, which are typically abundant in valid MaSp repeat sequences. In contrast, sequences containing non-repeat regions exhibited almost null densities of these characteristic motifs. Among the 100 generated sequences, 68% included non-repeat prefixes or suffixes, appearing at the beginning or end of the sequences, respectively (Figure 9). This finding suggests that, without the initial fine-tuning phase, the pretrained model retains its original behavior, failing to specialize in generating only the repetitive core region of MaSp sequences.

To further investigate these discrepancies, we compare the structural predictions of sequences generated by a model trained with only one fine-tuning stage against those produced by a model trained with both levels of fine-tuning. Figure 9 illustrates the molecular configurations of the generated sequences, revealing structural differences arising from the absence of MaSp-specific fine-tuning. Structural predictions were obtained using Omegafold (Wu et al., 2022), providing insights into the conformational tendencies of the sequences. Notably, the generated sequences without level 1 fine-tuning exhibited helical structures, which are typically associated with non repeat regions like terminal domains (NTD/CTD). These unwanted elements further highlight the necessity of level 1 fine-tuning in ensuring the correct generation of MaSp repeats.

**Impact on Property Prediction**

Here, we analyze the impact of skipping Level 1 fine-tuning on the reverse task of mechanical property estimation. Using the same test set, we prompted the model to predict mechanical properties from sequence inputs and compared the predicted values against their ground truth counterparts. To assess performance, we compared trend similarity metrics with and without Level 1 fine-tuning. When the model was trained

Table 3: Comparison of generated and reference properties without level 1 finetuning

| Metric | Mean Value |
|---|---|
| Pearson Correlation ($r$) | 0.3628 |
| Spearman Correlation ($\rho$) | 0.3301 |
| Mean Absolute Error (MAE) | 0.2911 |
| Root Mean Square Error (RMSE) | 0.3447 |
| R square ($r^2$) | -2.7196 |
| Cosine Similarity | 0.8255 |

without the 6,000 MaSp repeat sequences used in Level 1, its ability to recover accurate property trends declined significantly, as shown in Table 3. The mean Pearson correlation dropped from 0.8884 to 0.3628, and the mean Spearman correlation fell from 0.8343 to 0.3301, indicating a considerable loss in both linear and rank-order consistency.

Prediction errors also increased substantially: MAE rose from 0.0861 to 0.2911, and RMSE increased from 0.1047 to 0.3447. Most notably, the $R^2$ score deteriorated from 0.6383 to $-2.7196$, suggesting that without Level 1 fine-tuning, the model performs worse than a simple mean-based predictor. Cosine similarity also decreased from 0.9827 to 0.8255, reflecting weakened directional agreement between predicted and reference property vectors.

These results highlight the essential role of Level 1 fine-tuning in learning robust sequence embeddings that generalize well to downstream property prediction. Its absence leads to diminished trend fidelity and overall performance degradation.

## 6 Conclusion and Future Work

This research presents a novel generative model based on GPT architecture, meticulously fine-tuned to leverage a dataset comprising 6,000 repeat regions derived from spider silk proteins. This initial training was subsequently enhanced through a secondary fine-tuning phase utilizing approximately 600 Major Ampullate Spidroin (MaSp) sequences, each annotated with well-characterized mechanical properties. Our innovative dual-level fine-tuning strategy has proven effective in producing synthetic sequences that incorporate essential structural motifs, such as the glycine-rich GGX repeats and poly-Ala stretches, which are fundamental to the remarkable extensibility and tensile strength exhibited by natural spider silk.

Detailed structural analyses of the generated sequences reveal a strong alignment with the anticipated secondary structure composition, such as $\alpha$-helical and $\beta$-strand conformations, which are critical for replicating the functional attributes of spider silk. Furthermore, comparisons between the mechanical properties predicted by the model and established reference data underscore the reliability and promise of this approach for designing spidroins with tailored mechanical properties. By enabling the precise generation of sequences tailored to specific performance criteria, this work lays a robust foundation for the sustainable production of synthetic spider silk materials. The platform that we have developed is both scalable and highly customizable, offering significant potential to transform material science and synthetic biology. Its implications extend beyond spider silk, paving the way for the creation of next-generation bioinspired materials with applications in diverse fields such as tissue engineering, textiles, and environmentally friendly composites.

Future work will encompass a comprehensive experimental validation process for the generated sequences, which will involve synthesizing these sequences and subjecting them to rigorous mechanical testing. This step is crucial to verify that the predicted properties align with the actual performance characteristics observed under controlled conditions. The synthesis process will aim to accurately replicate the molecular structures proposed by the generative model, while the mechanical testing will evaluate key parameters such as tensile strength, extensibility, and toughness to ensure that the sequences meet the anticipated functional benchmarks.

A key direction for future work involves enhancing the generative model by integrating significantly larger and more diverse datasets to improve both predictive accuracy and generalizability. Scaling up the training data

will allow the model to better capture the complex, non-linear relationships between sequence composition and resulting mechanical properties, thereby refining its capacity to generate optimized MaSp repeat designs. While the current methodology demonstrates strong performance, it was specifically tailored to a relatively small, high-quality dataset.

To extend this framework, we plan to expand the spidroin-to-mechanical-properties dataset by leveraging a large corpus of unlabelled spidroin sequences available through the Spider Silkome resource (Arakawa et al., 2022). These sequences will be annotated with experimentally or computationally derived mechanical properties in future work, enabling their integration into the training pipeline. This expanded dataset will encompass a broader diversity of spidroin variants and their corresponding mechanical attributes, supporting the development of more robust sequence-to-function mappings and facilitating the design of advanced silk-based materials with tailored performance characteristics.

### Software and Data

The implementation of our model inference pipeline is available at: `https://github.com/Anon-GitAI/spiderfiber-demo`. The repository includes the necessary scripts, configuration files, and usage instructions. All relevant datasets used in this study are provided in the `Data` folder within the repository.

### Acknowledgments

This work was supported by grants from the SciLifeLab & Wallenberg Data Driven Life Science Program (KAW 2020.0239) and by the Wallenberg AI, Autonomous Systems and Software Program (WASP) funded by the Knut and Alice Wallenberg Foundation: WASP-DDLS 22:035 to AR and HK, KAW 2023.0331 and KAW 2017.0003 to AR, and by the National Bioinformatics Infrastructure Sweden (NBIS) at SciLifeLab. Support was also received by AR from FORMAS (2023-01313), Olle Engkvist stiftelse (233-0334), and the Swedish Research council (2024-02919), and by HK from the Swedish e-Science Research Centre (SeRC).

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

# A  Appendix

## A.1  Distillation Data Acquisition

Protein sequences were retrieved from the UniProt Knowledgebase (UniProtKB) using the UniProt REST API. To obtain high-quality sequences, we filtered entries based on taxonomy ID 6893 and annotation scores of 2, 3, 4, or 5. The following query was used to download the dataset in FASTA format:

```
https://rest.uniprot.org/uniprotkb/stream?compressed=true&format=fasta&
query=(taxonomy_id:6893) AND (annotation_score:2 OR annotation_score:3 OR
    annotation_score:4 OR annotation_score:5)
```

The dataset comprises protein sequences specific to the selected taxonomy, ensuring relevance for further computational analyses.

## A.2 Choice of Physiochemical Attributes for Evaluation

In current study we have used sequence length, molecular weight, instability index, and isoelectric point as physiochemical attributes to evaluate the quality of generated MaSp repeats in various contexts.

Molecular weight and sequence length are widely used as coarse-grained indicators of biological plausibility in protein design. Empirically, naturally occurring major ampullate spidroin (MaSp) proteins fall within a narrow range of lengths and molecular weights, reflecting functional and structural constraints (Yang et al., 2016). For instance, MaSp2 proteins often exhibit high molecular weights due to their repetitive sequences, which are crucial for their mechanical properties (Sino Biological, 2024). Deviations from these ranges can indicate sequences that are unlikely to fold correctly or be expressed efficiently, as proper folding is essential for protein functionality .

The instability index, introduced by (Guruprasad et al., 1990) Guruprasad et al. (1990), predicts the in vivo stability of a protein based on its dipeptide composition. In the context of MaSp repeats, evaluating the instability index helps in predicting the stability of the designed sequences, which is crucial for their functional expression and potential applications.

The isoelectric point is the pH at which a protein carries no net electrical charge. It significantly influences a protein's solubility, stability, and interaction with other molecules. Understanding the pI of MaSp repeat sequences aids in anticipating their behavior under different pH conditions, which is vital for their purification and application processes (Audain et al., 2016).

In summary, sequence length, molecular weight, instability index and isoelectric point are critical parameters in the evaluation of MaSp repeat sequences, providing insights into their physical constraints, stability and functional suitability in various applications.

## A.3 Standard Amino Acids with their Background Frequency & KL Divergence for MaSp Repeats

Like human language, protein sequences can be represented as strings of letters, where the protein alphabet consists of 20 standard amino acids (AAs), excluding rare and unconventional ones. Similarly, naturally evolved proteins are composed of modular elements with slight variations, which can be rearranged and assembled hierarchically. In this analogy, common protein motifs and domains—fundamental functional units of proteins—are akin to words, phrases, and sentences in human language (Ofer et al., 2021).

The twenty amino acids (that make up proteins)each have assigned to them both three-letter (can be upper or lower case) and one-letter codes (upper case). This makes it quicker and easier for notation purposes and are worth learning. Table 4 gives these notations.

For evaluation purposes, we established a background amino acid frequency distribution based on the training dataset of MaSp repeats (6K instance which were used in Stage 2). This distribution represents the mean occurrence of each amino acid across all sequences in the dataset. The calculated background frequencies are shown in Table 4.

This frequency distribution serves as a reference baseline for KL divergence calculations, enabling a quantitative comparison between generated sequences and naturally occurring MaSp repeats.

To assess sequence similarity, we calculate the Kullback-Leibler (KL) divergence between the amino acid composition of generated sequences and the background distribution of MaSp repeats. The background frequencies of amino acids were derived from a dataset of 6,000 MaSp repeat sequences, providing a reference probability distribution. Given a sequence $S$, its amino acid probability distribution $P$ is compared against the background distribution $Q$ using:

$$D_{KL}(P||Q) = \sum_i P(i) \log \frac{P(i)}{Q(i)} \tag{2}$$

where $P(i)$ and $Q(i)$ represent the probabilities of amino acid $i$ in the generated sequence and background dataset, respectively. This metric quantifies the deviation of generated sequences from natural MaSp repeats, aiding in model evaluation.

Table 4: Amino Acids, Their Codes, and Background Frequency Distribution for MaSp Repeats

| Amino Acid Name | 3-Letter Code | 1-Letter Code | Background Frequency |
|---|---|---|---|
| Alanine | Ala | A | 0.2232 |
| Arginine | Arg | R | 0.0129 |
| Asparagine | Asn | N | 0.0070 |
| Aspartic Acid | Asp | D | 0.0078 |
| Cysteine | Cys | C | 0.0002 |
| Glutamine | Gln | Q | 0.0850 |
| Glutamic Acid | Glu | E | 0.0069 |
| Glycine | Gly | G | 0.3766 |
| Histidine | His | H | 0.0003 |
| Isoleucine | Ile | I | 0.0050 |
| Leucine | Leu | L | 0.0138 |
| Lysine | Lys | K | 0.0017 |
| Methionine | Met | M | 0.0014 |
| Phenylalanine | Phe | F | 0.0038 |
| Proline | Pro | P | 0.0788 |
| Serine | Ser | S | 0.1004 |
| Threonine | Thr | T | 0.0123 |
| Tryptophan | Trp | W | 0.0002 |
| Tyrosine | Tyr | Y | 0.0485 |
| Valine | Val | V | 0.0141 |

