# OpenReview forum: "Customizing Spider Silk: Generative Models with Mechanical Property Conditioning for Protein Engineering"
_TMLR — Accepted by TMLR_

### Review · Reviewer_EX2U · 2025-05-01

**Summary Of Contributions:**

This paper presents a model for both forward prediction and inverse generation of sequence–property of a key ingredient in spider silks, specifically, MaSps repeat regions and mechanical properties. The model is developed by distilling a pre-trained protein language model, ProGPT2, and fine-tuning it in two stages. Experiments evaluate the plausibility, novelty, and accuracy of the model’s (conditional) generation, as well as the importance of distillation and fine-tuning (through ablation studies).

**Audience:**

Yes

**Broader Impact Concerns:**

None.

**Claims And Evidence:**

Yes

**Requested Changes:**

Elaboration or clarification
- (Chemical/biological) theory behind the role of MaSp (and its repeat regions) in determining mechanical properties could be better elaborated in Sec. 2.1.
  - A related question: In level 1 fine-tuning, the first and last amino acid residues are removed, but could they have interactions with the MaSp repeat regions? That is, changing these residues changes the correlation between MaSp sequence and property.
- The role of training parameters (T, α, etc.) used in Stage 1 should be explained, at least qualitatively.

Experiments and evaluation
- Physiochemical attributes such as molecular weight and sequence length are used to assess biological plausibility. Are there theoretical or empirical justifications that these attributes are related to plausibility?
- Sec. 5.1.1 compares the generated and natural sequences that correspond to the same mechanical properties, but this comparison is only meaningful if the sequence–property mapping is one-to-one. This comparison should be justified.
- In Sec. 5.2, Pearson r ~ 0.39 corresponds to R2~0.15, which is too low. Since the goal is assess “overall … trends”, perhaps Spearman r could be a useful metric to try.

Minor issues
- There are some clarity issues:
  - Sec. 4.3, “synthesized” is misleading. Should it be “generated”?
  - Typos, e.g., “row vise analysis” on Page 12.
  - Some terms should be unified, e.g., “repeat region” or “repetitive region”.
- In the main contributions, “practical contribution to sustainable…” seems overclaimed, given the model performance and lack of practical demonstration.

**Strengths And Weaknesses:**

Strengths
- The paper is well-written, clear and easy to follow.
- The components of proposed methodology are well motivated and tested in comprehensive ablation studies.

Weaknesses
- The performance of mechanical property prediction/design is poor (Sec. 5.2).
- The theory on MaSp could be better elaborated (see below).

---

> ### Author Response · Authors · 2025-06-05
> **Improved MaSp discussion and added Spearman correlation in to the evaluation set**
>
> ### Weaknesses
> 1.  We have now implemented cross validation in Satge 3 of the pipeline. This has improved the system performance significantly.
> 2. We have extended MaSp theory in the literature review Sec. 2.1.
>
> ### Elaboration or clarification
> 1. We agree that the chemical and biological theory behind MaSp and its repeat regions could be better elaborated. We have included a more detailed discussion Section 2.1 to include a clearer explanation of how the repeat region contribute to mechanical properties.
> 2. The reviewer raises an interesting point, which is to what degree the terminal domains contribute to the fibers’ mechanical properties. The terminal domains are highly conserved and shared between different spidroins that give rise to fibers of very diverse mechanical properties. This points to that they are primarily important for the storage and polymerization mechanism of the spidroins, not the mechanical properties. Both terminal domains have been extensively studied in the past (Andersson et al., 2014; Askarieh et al., 2010; Eisoldt et al., 2012; Gao et al., 2013; Hagn et al., 2010; Hagn et al., 2011; Kronqvist et al., 2014; Landreh et al., 2010; Otikovs et al., 2015; Sede et al., 2022; Wang et al., 2014), and to our knowledge, interactions between the terminal domains and the repeat region have not been described. Furthermore, currently available data and models of the fibers’ structure-function relationship point to the fact that it is the repetitive region of the spidroins that determines the mechanical properties of the fibers (Anton et al., 2017; Bowen et al., 2018; Gatesy et al., 2001; Gosline et al., 1999; Keten & Buehler, 2010; Kono et al., 2021; Nakamura et al., 2023; Nova et al., 2010; Rising et al., 2005; Schmuck et al., 2023; Teule et al., 2012; Vollrath & Knight, 2001; Yarger et al., 2018). Hence, in the current manuscript, our intention was to isolate the repeat region, which is known to primarily drive the mechanical behavior of MaSp proteins. While we cannot exclude that the terminal domains may have some interaction with the repetitive region, our design choice reflects a focus on the core motifs most directly linked to material properties. We will clarify this rationale in the revised manuscript.
> 3. Thank you for pointing this out. We have added a qualitative explanation of the training parameters used in Stage 1—such as temperature (T) and scaling factor (α)—to clarify their role in controlling sequence diversity and gradient scaling, respectively. This will help readers better understand their impact on the generation process.
>
> ### Experiments and evaluation
> 1. Thank you for raising this important question. Molecular weight and sequence length are widely used as coarse-grained indicators of biological plausibility in protein design. Empirically, naturally occurring MaSp proteins fall within a narrow range of lengths and molecular weights, which reflects functional and structural constraints. The range varies for each MaSp subtype. Deviations from these ranges often indicate sequences that are unlikely to fold correctly or be expressed efficiently. We have added the rationale behind selection of the physiochemical attributes in Appendix A.2.
> 2. We agree that the sequence–property relationship is not strictly one-to-one. Our comparison in Section 5.1.1 is based on the assumption that similar mechanical properties can arise from sequences with shared structural or motif-level features, particularly within the MaSp repeat regions. The goal was to qualitatively assess whether the generated sequences resemble natural ones associated with similar properties. We have now clarifed this assumption in the updated manuscript.
> 3. With the inclusion of cross-validation at stage 3 and the resulting increase in the test set size, we now observe much improved comparison results. The Pearson correlation has significantly improved, changing from 0.39 to 0.88. As suggested, we’ve also added Spearman correlation to our evaluation. The updated comparison results are presented in Table 1.
>
> ### Minor issues
> * Thank you for pointing out these clarity and consistency issues. Additionally, we have revised the statement on “practical contribution to sustainable biomaterials” as “potential solution for the synthesis of sustainable biomaterials” to more accurately reflect the current scope and limitations of our work.

---

### Review · Reviewer_V9sT · 2025-05-07

**Summary Of Contributions:**

This manuscript proposes a multi-stage workflow for fine-tuning a pre-trained protein language model (PLM) on the repetitive regions of spider silk proteins (MaSps). In the first stage, a smaller version of a generalist PLM (ProtGPT) is distilled on ~100k spider protein sequences. This model is then fine-tuned on ~6k MaSp sequences to capture their characteristic sequence motifs. Finally, the model is fine-tuned again on ~600 property-annotated MaSp sequences, which are augmented by task and property tokens to enable conditional sampling and property estimation.

**Audience:**

Yes

**Broader Impact Concerns:**

I have no concerns about the negative ethical implications of this work.

**Claims And Evidence:**

Yes

**Requested Changes:**

- It would be great to incorporate more comparisons and baselines into the results section of the paper, e.g., including sequences from the base ProtGPT model in the property comparisons in Figure 3 or comparing the method to existing approaches for mechanical property prediction and spindroin sequence design listed in Section 2.3.
- It would also be helpful to improve the robustness of the experimental evaluation itself, either by using larger test sets or performing bootstrapping or k-fold cross-validation over the test set.
- I found that there was some overlap in content between Sections 3, 4, and 5. I believe that parts of it could be condensed (or moved to the appendix) to create a more streamlined description of the methodology and experimental results.
- I also believe that some of the claims should be toned down. For instance, the Training Process paragraph on page 7 describes dropout, weight decay, and early stopping as "advanced optimization techniques". Similarly, top k model checkpointing is described as a "sophisticated checkpointing system". In my opinion, these are standard techniques and should not be portrayed as advanced methodological achievements.

**Strengths And Weaknesses:**

### Strengths
- The manuscript provides an excellent introduction to the subject matter and successfully conveys the importance of the topic.
- The manuscript provides a clear, high-level overview of the proposed multi-level fine-tuning methodology and the motivation behind it.

### Weaknesses
- If I understand correctly, the test sets used in Sections 5.1.1 and 5.1.2 consist of 20 and 7 sequences, respectively, without any kind of bootstrapping. This is extremely small, and I find it difficult to draw strong conclusions about model performance based on such small sample sizes.
- The paper does not include any baseline methods and only presents a limited set of ablations for the proposed approach.
- As far as I could tell, no code is provided to reproduce the empirical results reported in the manuscript. In the absence of an accompanying codebase, the description of the methodology in the Training Process paragraphs on pages 6, 7 and 8 seems incomplete, e.g. listing LoRA hyperparameters without defining the corresponding algorithm or providing an appropriate reference.

---

> ### Author Response · Authors · 2025-06-05
> **Implemented K-fold cross validation on Stage 3 (Level 2 fine tuning). This has increased the number of test instances from 20 to 185 and improved the overall system evaluation.**
>
> ### Weekness
> 1. Thank you for pointing this out. We agree that the test set sizes in Sections 5.1.1 and 5.1.2 are small and acknowledge the limitations this imposes on the robustness of the performance evaluation. To address this, we have now performed bootstrapping to obtain more reliable performance estimates and updated the results accordingly in the revised manuscript.
> 2. We would like to emphasize that the proposed approach is, to the best of our knowledge, the first of its kind for this specific task. Due to the absence of existing methods that directly address the same problem setup, there are no established baselines for a direct comparison.
> 3. Regarding the availability of the code, we have open-source code online, but we can't provide the link here for anonymity. We will provide the link to the code in our paper after the decision. Additionally, we used the standard LoRA implementation provided in Hugging Face’s PEFT library and will clarify this with a citation and brief explanation in the methodology section.
>
> ### Requested Changes:
> 1. The sequences in Figure 3 include only repeat regions of MaSp proteins, which exhibit distinct mechanical properties. ProtGPT2 generates general protein sequences, and existing methods discussed in Section 2.3 also focus on  MaSp proteins, including one of the terminal domains, not exclusively on the repeat region of MaSp. We clarify this distinction in the manuscript section 3.
> 2. Thank you for the suggestion. We are now employing bootstrapping to better assess model performance given the limited dataset size. The training procedure and results are updated in the revised manuscript.
> 3. Thank you for flagging the content overlap—Sections 3–5, We have removed some of the overlap and updated the manuscript.
> 4. We appreciate your feedback on the use of terminology. We have revised the language describing dropout, weight decay, early stopping, and checkpointing to accurately reflect their standard and widely adopted status in the field.

---

> > ### Comment · Reviewer_V9sT · 2025-06-10
> >
> > Thank you for the concise response and changes to the manuscript. I believe that they strengthen the paper and have adjusted my Claims and Evidence assessment in response. Please find additional comments below:
> >
> > **Regarding Weakness 2**, I understand that the proposed approach is the first method to enable joint sequence modeling and mechanical property prediction. However, is there a reason why each of these capabilities cannot be compared to the respective baseline performance of the existing methods [1] and [2] that are mentioned in Section 2.3? Even though [2] focuses on generating full sequences--- including the terminal domains---I don't see why it would not be possible to compare the subsequences that correspond to the MaSp repeat regions.
> >
> > **Regarding Weakness 3**: I believe that it would generally be beneficial to share an anonymized version of the code, either as supplementary material or via services such as https://anonymous.4open.science/.
> >
> > ---
> > ### References
> > [1] Yoonjung Kim, Taeyoung Yoon, Woo B Park, and Sungsoo Na. Predicting mechanical properties of silk from its amino acid sequences via machine learning. Journal of the Mechanical Behavior of Biomedical Materials, 140:105739, 2023
> >
> > [2] Wei Lu, David L Kaplan, and Markus J Buehler. Generative modeling, design, and analysis of spider silk protein sequences for enhanced mechanical properties. Advanced Functional Materials, 34(11):2311324, 2024.

---

> > > ### Author Response · Authors · 2025-06-12
> > > **Baseline comparison and code sharing**
> > >
> > > ## Regarding Weakness 2: Baseline Comparison
> > >
> > > - **Kim et al.** developed predictive machine learning models to estimate silk mechanical properties from full-length amino acid sequences (not only for MaSP). However, their approach does not support generative design or customizable sequence outputs. In contrast, our framework enables both **generation** and **prediction** of MaSp repeat regions, and is trained specifically on curated repeat-level datasets, making it structurally and functionally distinct from Kim et al.’s full-sequence regression approach.
> > >
> > > - **Lu et al.** introduced a generative large-language model (**SilkomeGPT**) trained on approximately 1,000 major ampullate spidroin (MaSp) sequences with associated fiber-level mechanical properties. The primary differences between the two models lie in input design and training data exposure:
> > >   - **SpiderGPT** uses only the **repeat region** as input.
> > >   - **SilkomeGPT** uses the **fuller sequence** (i.e., `[NTD+] repeat [+CTD]`).
> > >
> > > Kindly note that a **forward task comparison is not feasible** due to fundamental differences in sequence generation strategies. However, recognizing the importance of baseline comparisons, we have evaluated both models on the **reverse task of mechanical property prediction** using the test set and updated the manuscript accordingly (see Section 5.2).
> > >
> > > Furthermore, SilkomeGPT was trained on the **entire labeled Silkome dataset, including the test instances** used for evaluation. In contrast, SpiderGPT was trained exclusively on **repeat regions**, and the test sequences were **unseen during training**.
> > >
> > >
> > >
> > > ## Regarding Weakness 3:  Reproducibility and Code Availability
> > >
> > > We initially planned to release a curated GitHub repository—with Hugging Face model checkpoints for each fine-tuning stage—after the review process, as sharing the full pipeline risks compromising anonymity.
> > >
> > > However, to support reproducibility and enable verification of model performance, we now provide a **anonymized implementation of our inference pipeline**. This includes example usage, dependency specifications, and essential scripts.
> > >
> > > You can access the repository here:
> > > [https://github.com/Anon-GitAI/spiderfiber-demo](https://github.com/Anon-GitAI/spiderfiber-demo)
> > >
> > > The repository contains all necessary configurations to reproduce the results reported in the manuscript, while maintaining author anonymity.

---

### Review · Reviewer_Hf5q · 2025-05-27

**Summary Of Contributions:**

**Summary Of Contributions**

*   The paper presents a **novel computational framework** designed to engineer protein sequences for spider silk, specifically focusing on the **repeat regions of major ampullate spidroins (MaSps)**. This framework aims to enable the **customization of mechanical properties** in synthetic spider silk materials. It addresses the challenge of limited data connecting MaSp repeat sequences directly to mechanical properties by employing a **multi-stage fine-tuning approach**.
*   A key contribution is the development of a **dual-purpose generative model**. This model is capable of two main tasks: **generating novel MaSp repeat sequences conditioned on desired mechanical properties** (the "forward task") and **predicting the mechanical properties for a given MaSp repeat sequence** (the "reverse task"). This flexibility supports both the design of new sequences and the analysis of existing ones.
*   The methodology involves distilling a large pre-trained protein language model, ProtGPT2, into a smaller, more efficient model called **SpiderGPT**. This distilled model undergoes two levels of fine-tuning using subsets of the Spider Silkome dataset. The first level tunes the model on 6,000 MaSp repeat sequences to learn their general patterns, while the second refines it using 572 sequences with known mechanical properties to establish sequence-property correlations. Ablation studies highlight the importance of both distillation for computational efficiency and the initial fine-tuning stage for ensuring the generation of valid MaSp repeat regions and improving property prediction accuracy.
*   A **comprehensive validation methodology** was employed to evaluate the generated sequences. This included sequence-level analysis of physicochemical properties (like molecular weight, instability index, isoelectric point, and amino acid distribution), assessment of key motif distributions (such as poly-Ala, GGX, YGQGG, and SV), structural prediction to check secondary structure composition (like α-helices and β-strands), and evaluation of mechanical property correlation.
*   The paper empirically demonstrates the model's effectiveness. Generated sequences were shown to be **biologically plausible**, closely matching the distributions of physicochemical attributes and secondary structures found in natural MaSp repeats. They also maintained distributions of key motifs similar to natural sequences. A BLAST-based novelty assessment indicated the model can generate diverse sequences while retaining the characteristic features of MaSps. For mechanical property prediction, the model achieved **statistically significant correlations** between predicted and reference properties, with a high cosine similarity of 0.9465 for trend curves, suggesting strong predictive performance. Although individual motif-property correlations remained weak, consistent with previous findings on natural silk, the model's overall property estimation accuracy was demonstrated.
*   The work provides a **practical and scalable approach** for designing synthetic spider silk proteins with tailored mechanical properties. This is presented as a significant contribution to **sustainable biomaterial development** with potential applications in diverse fields such as medicine, textiles, and engineering.

**Audience:**

Yes

**Broader Impact Concerns:**

No concerns as such.

**Claims And Evidence:**

Yes

**Requested Changes:**

List of proposed adjustments to the submission:

**Critical for Acceptance:**

*   **Comprehensive Literature review:** The important multi-stage method discussed over here (in section 2.4) and its major contributions are interesting, but it has been discussed in previous publications in different scientific areas. One of the recent publications submitted in this year's ICLR implements a multi-stage model for audio sciences highly similar to the current work. Described in major contribution as "two-step post-training" in (https://arxiv.org/abs/2408.09269), kindly include such works and others that you may find appropriate to your work.

*   **Schematic of model overview** Kindly include a proper figure for the architecture that gives the overview of the model, input, outputs and conditions. Also, include an explanation of your data (using mathematical notations) and proper set notation.

*   **Important proof of propositions** While you are using a generative model, kindly include a note as to why you think the generated output is correct and preserves the physical attributes of the input dataset (i.e if the input dataset follows some material properties, then why will the output follow such properties). It is important to address this while you are developing generative models for scientific applications. Kindly refer to the proposition from A.1.1 (https://arxiv.org/pdf/2408.07213) and section A.1-A.2 (https://arxiv.org/pdf/2307.02707). (Kindly do not get perplexed by the mathematics, the authors are only requested to understand the importance of these propositions while using a generative model of any kind for a physical system like the one discussed in this paper).

*   **Comprehensive Experimental Validation:** The most significant adjustment critical for acceptance would be the **initiation or inclusion of experimental validation** of the generated sequences. The paper currently states this as future work. While the computational validation is thorough, demonstrating the practical utility of the framework by synthesizing and mechanically testing at least a representative sample of generated sequences to verify their predicted properties is often essential for applied biomaterial design research. This step is crucial to verify that the predicted properties align with actual performance characteristics.

**Strengthens the Work:**

*   **More Explicit Discussion of Data Limitations:** While the multi-stage fine-tuning is presented as a solution to the challenge of the "extremely small" target dataset, the paper could benefit from a more in-depth discussion of the **potential impact of this inherent data scarcity** on the model's limits in terms of the diversity of properties it can condition on, the precision of its predictions, and its generalizability to property ranges significantly outside those represented in the 592-instance dataset.
*   **Further Elaboration on Weak Motif-Property Correlations:** The finding that individual motifs show only weak correlations with mechanical properties, consistent with previous research, is mentioned. Expanding the discussion on the **implications of this inherent complexity** of spider silk mechanics for computational design would be valuable. This could reinforce why a holistic, data-driven model capturing complex interplay is necessary, rather than relying on simple motif engineering.
*   **Addressing Limitations in Standard Structural Prediction:** The paper notes that standard secondary structure prediction methods are "less reliable for structural proteins like MaSp repeats". While using these methods is standard practice, a brief discussion of alternative, more specialised structural prediction methods or a clearer acknowledgement of the **uncertainty associated with the structural predictions** when interpreting results would strengthen this section.
*   **Quantification or Mitigation of Validation Sensitivity:** The authors mention that variations in BLAST comparison metrics might be due to the "sensitivity of the evaluation process to the chosen dataset". Providing more detail on **how the reference BLAST set was curated** to minimise bias or attempting to quantify this sensitivity through comparisons against multiple different reference sets could strengthen the novelty assessment.
*   **Improving or Discussing Moderate Linear Property Correlation:** While the cosine similarity is high (0.9465) indicating good trend capture, the Pearson correlation coefficient (r) for predicted vs. reference properties is described as a "moderate positive correlation" (0.3911). While statistically significant, discussing potential avenues for **improving the linear accuracy** of property prediction or explaining the potential reasons for this moderate correlation (e.g., data noise, inherent system complexity) could enhance the paper.
*   **Strengthened Commitment to Future Data Expansion:** Reiterate the importance of acquiring **significantly larger and more diverse datasets**. While stated as future work, providing more concrete plans or strategies for this data acquisition and integration would underscore a clear path towards overcoming current limitations and improving model accuracy and generalizability.

**Strengths And Weaknesses:**

**Strengths**

*   **Novel Computational Framework:** The paper introduces a **novel computational framework** for engineering protein sequences, specifically focusing on the repeat regions of major ampullate spidroins (MaSps) to customise mechanical properties. This framework addresses the challenge posed by limited data connecting sequences directly to mechanical properties.
*   **Dual Functionality:** The developed model serves a **dual purpose**, capable of both **generating new MaSp repeat sequences conditioned on desired mechanical properties** (the "forward task") and **predicting mechanical properties for a given MaSp repeat sequence** (the "reverse task"). This offers significant flexibility for both design and analysis.
*   **Efficient Model Development:** The approach involves **distilling a large pre-trained protein language model (ProtGPT2) into a smaller, more efficient model called SpiderGPT**. Ablation studies confirm that this distillation significantly **reduces model size and increases inference speed** (six-fold faster) with only minimal performance trade-offs in perplexity levels.
*   **Effective Multi-Level Fine-Tuning:** The methodology employs a **multi-stage fine-tuning process** (first on 6,000 general MaSp repeats, then on 572 with known mechanical properties). This staged approach is crucial, as ablation studies show that skipping the first fine-tuning level leads to the **generation of biologically implausible sequences** containing non-repeat regions and significantly **reduces accuracy in property prediction**.
*   **Comprehensive Validation:** A **comprehensive validation methodology** is used, including sequence-level analysis (physicochemical properties, motif distribution), structural prediction, and mechanical property correlation.
*   **Biologically Plausible and Novel Generation:** The generated sequences were shown to be **biologically plausible**, matching the distributions of physicochemical attributes, key motifs (poly-Ala, GGX, YGQGG, SV), and predicted secondary structures found in natural MaSp repeats. BLAST analysis indicated the model can **generate novel sequences** that are distinct from existing ones while retaining characteristic MaSp features.
*   **Demonstrated Predictive Capability:** The model demonstrated the capability to estimate mechanical properties, achieving a **statistically significant correlation** between predicted and reference properties on a test set. A **high cosine similarity (0.9465)** suggests strong alignment in capturing the *trends* of mechanical properties.
*   **Practical for Biomaterial Design:** The work provides a **practical and scalable approach** for designing synthetic spider silk proteins with tailored mechanical properties, contributing to **sustainable biomaterial development**.

**Elements Requiring Attention (Potential Weaknesses / Limitations Noted):**

*   **Literature review:** The literature review seems to be highly restricted to the niche field of fibre studies. While the concepts of deep learning which have been used in the current work, have already been worked upon extensively in the past.
*   **Limited Data:** The authors explicitly state that the target dataset mapping MaSp repeats to mechanical properties is **"extremely small"** (592 instances). While the multi-stage fine-tuning is designed to mitigate this, the inherent data scarcity remains a challenge.
*   **Weak Correlation of Individual Motifs:** Consistent with previous studies on natural silk, the analysis of individual motif occurrences and their correlation with mechanical properties in the *generated sequences* shows **relatively low or weak correlation values**. This reinforces the idea that silk mechanics result from a complex interplay of multiple motifs and structural contexts, but it means the model's direct mapping of single motifs to properties is limited.
*   **Sensitivity in Validation:** The authors note that variations observed in the BLAST comparison metrics (Figure 6) might be due to the **sensitivity of the evaluation process to the chosen dataset** of reference sequences.
*   **Moderate Linear Correlation in Property Prediction:** Although the cosine similarity is high (0.9465) indicating good trend prediction, the **Pearson correlation coefficient (r) for predicted vs. reference mechanical properties is 0.3911**, described as a "moderate positive correlation". While significant in context, it suggests the model's ability to precisely predict absolute property values might be less strong than its ability to capture relative trends.
*   **Reliance on Standard Structural Prediction Tools:** The paper uses standard secondary structure prediction methods (like those based on DSSP/Chou-Fasman) but notes that these are **"less reliable for structural proteins like MaSp repeats"** compared to globular proteins.
*   **Experimental Validation Pending:** A key piece of future work is the **comprehensive experimental validation** of the generated sequences by synthesizing and mechanically testing them. The current validation is primarily computational; experimental verification is necessary to fully confirm the practical utility and predicted properties of the sequences.
*   **Need for Larger Datasets:** Future work mentions the plan to **integrate significantly larger and more diverse datasets** to improve the predictive accuracy and generalizability of the model, implying that the current dataset size is a limitation to achieving higher performance.

---

> ### Author Response · Authors · 2025-06-05
>
> ### Elements Requiring Attention (Potential Weaknesses / Limitations)
>
> 1. Sections 2.2 and 2.3 discuss prior work on protein design via deep learning. While DL methods are well-studied broadly, their application to spidroins remains underexplored. We've expanded the review to acknowledge foundational DL work.
> 2. Data scarcity is a major challenge. Natural spidroin sequences with mechanical property annotations are difficult to obtain due to complex experiments. Despite this, we believe our study makes a meaningful contribution toward learning-based design.
> 3. Our model captures non-linear interactions at the sequence level rather than relying on isolated motifs—a key strength noted at the end of Section 4.4.
> 4. BLAST alignment can vary with minor sequence changes. We've transparently discussed this limitation as it applies uniformly across our dataset.
> 5. The sequence-to-property mapping is one-to-many but roughly monotonic. Our focus is on relative trends. The updated setup shows high cosine similarity (0.9827) and Pearson \( r = 0.88 \), indicating strong trend consistency (see Section 5.2).
> 6. We use available tools for large-scale secondary structure analysis consistently across sequences to enable comparative insights, acknowledging their limitations.
> 7. As mentioned in Section 6, experimental validation is part of future work. We are collaborating with biologists to synthesize predicted fibers and test them in the lab.
> 8. Larger datasets would improve generalizability. Our curated set is currently the best available with reliable labels. It enables meaningful insights and we position this work as a foundational step.
>
> ### Critical for Acceptance
>
> 1. We recognize the value in drawing parallels to similar methodologies in other fields. We have updated Section 2.4 to include this and other relevant works and clarify the novelty of our approach in the context of MaSp protein design.
> 2. Our methodology builds on the pretrained ProtGPT2 model, whose architecture is publicly available. However, we agree that including the architecture of the distilled model is valuable, and we will provide a schematic in the supplementary material. The training data consists of spidroin sequences with task tokens, and its mathematical formulation is provided in Section 3.3 (Equations 2 and 3) using set notation.
> 3. Our work focuses on generating spidroin sequences conditioned on mechanical properties, where the sequence-to-property relationship is biological and inherently one-to-many, unlike the structured physical systems discussed in the cited works. However, we agree on the importance of preserving relevant physical attributes. To ensure this, we evaluate unconditional as well as condition generation on property tokens and validate the outputs using trend alignment (e.g., cosine similarity) and distributional analyses. The same has been mentioned in paper, in section 4.3 and 5.1.1.
> 4. We fully agree that experimental validation is critical for applied biomaterial design. However, due to the high cost and time required for synthesis and mechanical testing, this step was beyond the current scope. Our goal in this work was to first establish a computational framework, which we hope can lay the foundation for the next steps of experimental collaborations.
>
> ### Strengthens the Work
>
> 1. The multi-stage training approach is designed to compensate for limited labeled data. We've already acknowledged dataset limitations and can clarify further if needed.
> 2. We emphasize the need for data-driven models to capture weak sequence–property correlations (see Sections 1 and 4.4).
> 3. We use standard secondary structure tools to enable consistent, large-scale analysis while being transparent about their limitations.
> 4. The reference BLAST set includes validated sequences from diverse MaSp subtypes with varied mechanical profiles (Section 5.1.2). We're open to clarifying this further.
> 5. Our goal is to model overall trends (e.g., high toughness/low modulus), not precise values. Cross-validation in Stage 3 has further improved performance.
> 6. We plan to expand our dataset using a large unlabelled spidroin dump, which we aim to annotate and integrate in future work (updated in Section 6).

---

> > ### Comment · Reviewer_Hf5q · 2025-06-19
> > **Thanks for the rebuttal**
> >
> > I acknowledge the rebuttal by the authors.

---

### Decision · Action_Editor_Te8t · 2025-07-07

**Recommendation:** Accept as is

**Audience:**

Yes

**Audience Explanation:**

The attempted task of generating proteins inspired by/based on spider silk is of high interest due to their properties. The paper advances the field using an interesting (though known) set of techniques.

**Claims And Evidence:**

Yes

**Claims Explanation:**

The paper proposes a framework to train generators capable of designing proteins inspired by/based on spider silk. In-silico experiments show that the designed structures follow the overall distribution of structural and biochemical features of known spider silk proteins. Furthermore, the designs are novel, exhibiting below 50-60% sequence similarity to known sequences.

The core contribution is a multi-level fine-tuning framework to train generators capable of designing novel and valid MaSp proteins. The framework results in promising & novel designs that match in distribution the core structural and biochemical features of known MaSps. Ablations further support that the introduced components of the pipeline such as distillation are important.

A key limitation of the work is the lack of experimental validation; any in-silico validation can only go as far.

All in all, the Reviewers have recommended accepting the paper and it is my pleasure to also recommend accepting the manuscript as is.